# Rescue of behavioral and electrophysiological phenotypes in a Pitt-Hopkins syndrome mouse model by genetic restoration of *Tcf4* expression

Hyojin Kim[1,2†], Eric B Gao[1,2], Adam Draper[1,2], Noah C Berens[3], Hanna Vihma[1,2], Xinyuan Zhang[4], Alexandra Higashi-Howard[4], Kimberly D Ritola[5‡], Jeremy M Simon[2,6,7], Andrew J Kennedy[4], Benjamin D Philpot[1,2,7]*

[1]Department of Cell Biology and Physiology, University of North Carolina at Chapel Hill, Chapel Hill, United States; [2]Neuroscience Center, University of North Carolina at Chapel Hill, Chapel Hill, United States; [3]Department of Biology, University of North Carolina at Chapel Hill, Chapel Hill, United States; [4]Department of Chemistry and Biochemistry, Bates College, Lewiston, United States; [5]HHMI Janelia Research Campus, Ashburn, United States; [6]Department of Genetics, University of North Carolina at Chapel Hill, Chapel Hill, United States; [7]Carolina Institute for Developmental Disabilities, University of North Carolina at Chapel Hil, Chapel Hill, United States

*For correspondence: bphilpot@med.unc.edu

Present address: †Life Edit Therapeutics, Morrisville, United States; ‡Neuroscience Center, University of North Carolina at Chapel Hill, Chapel Hill, United States

**Abstract** Pitt-Hopkins syndrome (PTHS) is a neurodevelopmental disorder caused by monoallelic mutation or deletion in the *transcription factor 4* (*TCF4*) gene. Individuals with PTHS typically present in the first year of life with developmental delay and exhibit intellectual disability, lack of speech, and motor incoordination. There are no effective treatments available for PTHS, but the root cause of the disorder, *TCF4* haploinsufficiency, suggests that it could be treated by normalizing *TCF4* gene expression. Here, we performed proof-of-concept viral gene therapy experiments using a conditional *Tcf4* mouse model of PTHS and found that postnatally reinstating *Tcf4* expression in neurons improved anxiety-like behavior, activity levels, innate behaviors, and memory. Postnatal reinstatement also partially corrected EEG abnormalities, which we characterized here for the first time, and the expression of key TCF4-regulated genes. Our results support a genetic normalization approach as a treatment strategy for PTHS, and possibly other TCF4-linked disorders.

## Editor's evaluation

The manuscript provides a proof of principle concept for rescue of a relatively common neurodevelopmental syndrome. By developing a novel Tcf4 conditional mouse model and demonstrating that PTHS phenotypes could be rescued by Tcf4 reinstatement during early postnatal development in particular cell types, the work sets the stage for future therapeutic efforts.

## Introduction

Pitt-Hopkins syndrome (PTHS) is a severe neurodevelopmental disorder, characterized by delay in motor function, lack of speech, stereotypies, sleep disorder, seizures, and intellectual disability. Other commonly reported features include constipation and hyperventilation (*Bedeschi et al., 2017*;

*Goodspeed et al., 2018*; *Zollino et al., 2019*). While PTHS is a lifelong disorder, there are currently no treatments for PTHS (*Zollino et al., 2019*). PTHS is caused by monoallelic mutation or deletion of *transcription factor 4* (*TCF4*), which is a member of the class I basic-helix-loop-helix (bHLH) group (*Amiel et al., 2007*; *Zweier et al., 2007*). PTHS-causing mutations typically impair the function of the bHLH domain, which is responsible for dimerizing with other bHLH proteins and for binding to Ephrussi box DNA elements to regulate transcription (*Dennis et al., 2019*; *Sepp et al., 2012*). Thus, targeting genes dysregulated by *TCF4* haploinsufficiency could potentially serve as a therapeutic intervention. However, hundreds to thousands of genes lie downstream of TCF4 (*Doostparast Torshizi et al., 2019*; *Forrest et al., 2013*; *Hill et al., 2017*; *Xia et al., 2018*), making it nearly impossible to find transcriptional modifiers to correct their expression levels. While targeting TCF4-impacted genes presents a fundamental conceptual challenge as a therapeutic approach, directly overcoming the core genetic defect underlying PTHS may offer a more effective treatment strategy.

In principle, PTHS phenotypes could be prevented or corrected by normalizing *TCF4* expression, with the degree of efficacy likely impacted by the age and specificity of the intervention. Convergent lines of evidence support the idea that the disorder can be treated, at least to a degree, throughout life. For example, studies in animal models of other single-gene neurodevelopmental disorders, including Rett and Angelman syndromes, have shown that normalizing expression of the disease-causing gene in postnatal life could provide therapeutic benefits (*Guy et al., 2007*; *Silva-Santos et al., 2015*). Therefore, the same might be true for PTHS. While synaptic defects have been observed in mouse models of PTHS, there is no evidence for disease-related neurodegeneration in PTHS individuals or mouse models (*Rannals et al., 2016*; *Thaxton et al., 2018*). Therefore, the observed synaptic defects could be reversible. In support of this idea, subtle upregulation of *Tcf4* expression by knocking down *Hdac2* has been shown to partially rescue learning and memory in adult PTHS model mice (*Kennedy et al., 2016*). Collectively, these observations indicate that PTHS might benefit from genetic normalization approaches to compensate for loss-of-function of TCF4 such as gene therapy, antisense oligonucleotides (ASOs), and small molecules.

A critical question that must be addressed prior to developing genetic normalization approaches for PTHS is whether behavioral phenotypes can be rescued if *TCF4* expression is restored during postnatal development. This question is particularly intriguing given observations that *TCF4/Tcf4* expression in the human/mouse brain peaks perinatally, before subsequently declining to basal levels that are sustained throughout adulthood (*Phan et al., 2020*; *Rannals et al., 2016*). Here, we leveraged a mouse model to establish the extent to which conditional reinstatement of *Tcf4* expression could rescue behavioral phenotypes in a mouse model of PTHS. We first validated our approach by demonstrating that embryonic pan-cellular reinstatement of *Tcf4* expression could fully prevent PTHS-associated phenotypes, while embryonic *Tcf4* reinstatement selectively in excitatory or inhibitory neurons led to rescue of only a subset of behavioral phenotypes. We then modeled viral gene therapy to show that postnatal reinstatement of *Tcf4* expression in neurons can fully or partially rescue behavioral and electrophysiological phenotypes in a mouse model of PTHS. Our results provide evidence that postnatal genetic normalization strategies offer an effective therapeutic intervention for PTHS.

## Results

### Pan-cellular embryonic reinstatement of *Tcf4* fully rescues behavioral phenotypes in a PTHS mouse model

We generated a conditional *Tcf4* reinstatement mouse model of PTHS (*Tcf4*$^{STOP/+}$) in which a transcriptional STOP cassette and GFP reporter, flanked by loxP sites, were inserted upstream of the basic Helix-Loop-Helix (bHLH) DNA binding domain in exon 18 of *Tcf4* (*Figure 1A*). To produce embryonic, pan-cellular reinstatement of *Tcf4*, we crossed *Tcf4*$^{STOP/+}$ mice to transgenic mice expressing Cre under the Actb promoter (*Jägle et al., 2007*). As predicted from our design, the levels of full-length *Tcf4*/TCF4 were reduced by approximately half in *Tcf4*$^{STOP/+}$ mouse brain compared to *Tcf4*$^{+/+}$ (wildtype control) mouse brain and fully normalized by crossing *Tcf4*$^{STOP/+}$ to *Actb-Cre*$^{+/-}$ mice (*Figure 1B*, *Tcf4*$^{+/+}$: 1.0 ± 0.09, n = 7; *Tcf4*$^{STOP/+}$: 0.61 ± 0.07, n = 7; *Tcf4*$^{STOP/+}$::*Actb-Cre*: 1.04 ± 0.09, n = 5, and *Figure 1C*, *Tcf4*$^{+/+}$: 1.0 ± 0.07, n = 7; *Tcf4*$^{STOP/+}$: 0.58 ± 0.05, n = 7; *Tcf4*$^{STOP/+}$::*Actb-Cre*: 1.07 ± 0.07, n = 5). We stained for the GFP reporter in sagittal brain sections from *Tcf4*$^{+/+}$, *Tcf4*$^{STOP/+}$, and *Tcf4*$^{STOP/+}$::*Actb-Cre* mice (*Figure 1—figure supplement 1A*). GFP signal, a proxy for presence of the STOP cassette, was

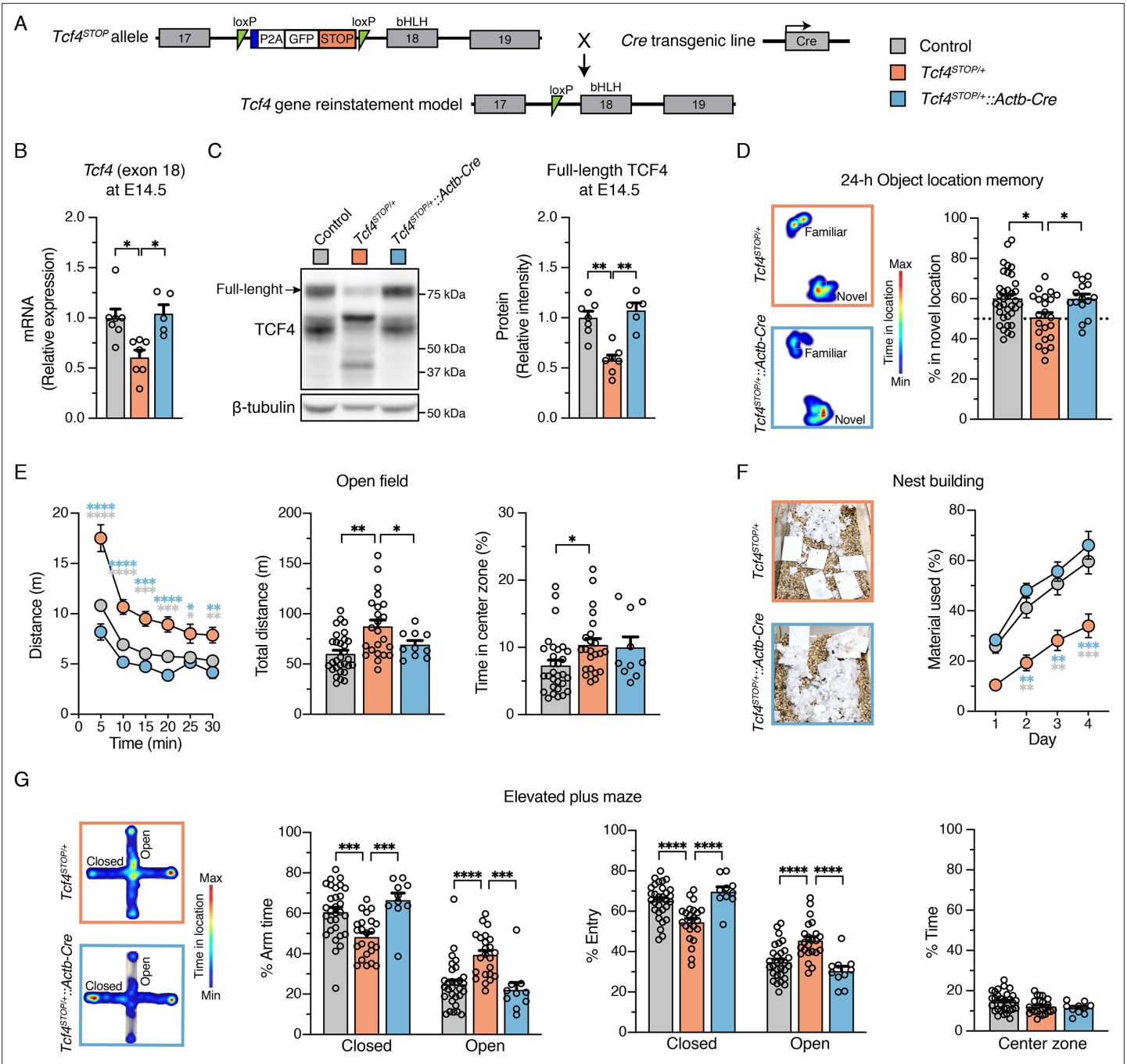

**Figure 1.** Embryonic, pan-cellular reinstatement of *Tcf4* fully rescues behavioral deficits in a mouse model of Pitt-Hopkins syndrome. (**A**) Schematic depicting a conditional Pitt-Hopkins syndrome mouse model in which expression of the bHLH region of *Tcf4* is prevented by the insertion of a loxP-P2A-GFP-STOP-loxP cassette into intron 17 of *Tcf4* (*Tcf4^STOP/+^*). Adenovirus splicing acceptor is shown by the blue box. Crossing *Tcf4^STOP/+^* mice with *Actb-Cre^+/−^* transgenic mice can produce mice with embryonic pan-cellular reinstatement of *Tcf4* expression (*Tcf4^STOP/+^::Actb-Cre*). (**B**) Relative *Tcf4* mRNA expression in embryonic brain lysates from *Tcf4^+/+^*, *Tcf4^STOP/+^*, and *Tcf4^STOP/+^::Actb-Cre*. The primers were designed to detect *Tcf4* exon 18. (**C**) Representative Western blot for TCF4 and β-tubulin loading control protein and quantification of relative intensity of TCF4 protein in embryonic brain lysates from *Tcf4^+/+^*, *Tcf4^STOP/+^*, and *Tcf4^STOP/+^::Actb-Cre* mice. The data were analyzed by one-way ANOVA followed by Bonferroni's *post hoc*. (**D**) Left panel: Heatmaps indicate time spent in proximity to one object located in the familiar position and the other object relocated to the novel position. Right panel: Percent time interacting with the novel location object (*Tcf4^+/+^*: n = 36, *Tcf4^STOP/+^*: n = 22, *Tcf4^STOP/+^::Actb-Cre*: n = 15). (**E**) Open field data. Left panel: Distance traveled per 5 min. Center panel: Total distance traveled for the 30 min testing period. Right panel: Percent time spent in the center zone (*Tcf4^+/+^*: n = 30, *Tcf4^STOP/+^*: n = 23, *Tcf4^STOP/+^::Actb-Cre*: n = 10). (**F**) Nest building data. Left panel: Representative images of nests built by *Tcf4^STOP/+^* and *Tcf4^STOP/+^::Actb-Cre* mice. Right panel: Percentage of nest material used during the 4 day nest building period (*Tcf4^+/+^*: n = 13, *Tcf4^STOP/+^*: n

*Figure 1 continued on next page*

*Figure 1 continued*

= 10, *Tcf4^{STOP/+}::Actb-Cre*: n = 5). (**G**) Elevated plus maze data. Left panel: Heatmaps reveal relative time spent on the elevated plus maze. Right panels: Percent time spent in the closed and open arms, percent of entries made into the closed and open arms, and percent time spent in the center zone (*Tcf4^{+/+}*: n = 30, *Tcf4^{STOP/+}*: n = 23, *Tcf4^{STOP/+}::Actb-Cre*: n = 10). Values are means ± SEM. *p < 0.05, **p < 0.005, ***p < 0.001, ****p < 0.0001.

The online version of this article includes the following source data and figure supplement(s) for figure 1:

**Source data 1.** Numerical data shown in *Figure 1*.

**Source data 2.** Numerical data shown in *Figure 2*.

**Figure supplement 1.** Body and brain weight analysis.

detected throughout the *Tcf4^{STOP/+}* mouse brain. By contrast, this signal was absent from *Tcf4^{+/+}* and *Tcf4^{STOP/+}::Actb-Cre* brain sections, indicating efficient excision of the GFP-STOP cassette and concomitant reinstatement of biallelic *Tcf4* expression in the *Tcf4^{STOP/+}::Actb-Cre* model mice (*Figure 1—figure supplement 1A*). To demonstrate the consequences of pan-cellular embryonic *Tcf4* reinstatement in PTHS model mice, we studied a variety of physiological functions and behaviors in control (*Tcf4^{+/+}* and *Tcf4^{+/+}::Actb-Cre*), PTHS model (*Tcf4^{STOP/+}*), and reinstatement model (*Tcf4^{STOP/+}::Actb-Cre*) mice. Male and female *Tcf4^{STOP/+}* mice had reduced body and brain weights, whereas *Tcf4^{STOP/+}::Actb-Cre* mice had similar body and brain weights to their littermate controls (*Figure 1—figure supplement 1B-C*). Reduced brain weight in *Tcf4^{STOP/+}* mice represents a phenotype with high face validity for human microcephaly (*Dupuis et al., 2015*; *Pulvers et al., 2010*; *Zhou et al., 2013*). Thus, our data suggest that embryonic reinstatement of *Tcf4* expression could prevent microcephaly in PTHS model mice. To study whether long-term memory deficits could be prevented, we examined object location memory by measuring interaction time of identical objects, with one object located in the familiar position and the other in a novel position. *Tcf4^{STOP/+}* interactions with objects located in the familiar and novel positions were of similar duration, suggesting the inability to remember the previously-presented location of the object and suggestive of long-term memory deficits (*Figure 1D*). *Tcf4^{STOP/+}::Actb-Cre* and control mice both interacted to a significantly greater extent with the object located in the novel position, suggesting recovery of normal long-term memory function subsequent to embryonic, pan-cellular *Tcf4* reinstatement (*Figure 1D*). We then assessed locomotor and exploration activity by the open field test and found that *Tcf4^{STOP/+}* mice showed increased activity and total distance travelled compared to *Tcf4^{STOP/+}::Actb-Cre* mice, which exhibited control levels of activity (*Figure 1E*). We also examined nest building, an innate, goal-directed behavior achieved by pulling, carrying, fraying, push digging, sorting, and fluffing of nest material (*Deacon, 2006*). *Tcf4^{STOP/+}* mice exhibited poor performance in the nest building task over the 4-day testing period, using roughly half the nest material used by control mice. This phenotype was completely rescued in the *Tcf4^{STOP/+}::Actb-Cre* model (*Figure 1F*). To assess anxiety-like behavior, we evaluated mice in the elevated plus maze task. We observed *Tcf4^{STOP/+}* mice to spend similar amounts of time in the closed and open arms, indicating an apparent low-anxiety phenotype. This behavioral feature also appeared to be normalized in *Tcf4^{STOP/+}::Actb-Cre* mice, spending proportionally more time in the closed arms (*Figure 1G*). Collectively, our results confirm that *Tcf4^{STOP/+}* mice exhibit physiological and behavioral phenotypes like those observed in other mouse models of PTHS (*Kennedy et al., 2016*; *Thaxton et al., 2018*), demonstrating the efficacy of the transcriptional STOP cassette in blocking TCF4 function. Moreover, these data show that *Tcf4* reinstatement upon embryonic Cre-mediated excision of the STOP cassette can fully prevent the emergence of physiological and behavioral deficits associated with *Tcf4* haploinsufficiency.

## *Tcf4* reinstatement in glutamatergic or GABAergic neurons rescues selective behavioral phenotypes in PTHS model mice

Viral-mediated gene delivery can target discrete cell types by promoter choice (*Deverman et al., 2018*), providing a capacity to adjust *Tcf4* expression in a cell type-specific manner. A previous anatomical study shows that TCF4 is present in excitatory and inhibitory neurons of the forebrain (*Kim et al., 2020*). Moreover, single-cell transcriptomic studies in the neonatal and adult mouse brain indicate that *Tcf4* transcript levels are higher in excitatory and inhibitory neurons than most other cell types (*Figure 2—figure supplement 1*; *Loo et al., 2019*; *Zeisel et al., 2015*). Thus, PTHS-associated pathologies might be effectively treated by preferentially reactivating *Tcf4* expression in excitatory and inhibitory neurons. To explore this possibility, and whether these broad neuronal subclasses

contribute to PTHS phenotypes in a modular or cooperative fashion (or both) in the case of *Tcf4* haploinsufficiency, we crossed *Tcf4^STOP/+^* mice to *Neurod6-Cre^+/-^* or *Gad2-Cre^+/-^* mice to reactivate *Tcf4* expression preferentially in forebrain glutamatergic neurons (*Tcf4^STOP/+^::Neurod6-Cre* mice) or GABAergic neurons (*Tcf4^STOP/+^::Gad2-Cre* mice) (*Figure 2A*). We then analyzed behavioral outcomes in these mice. In the open-field test, we found that *Tcf4^STOP/+^::Neurod6-Cre* mice exhibited significantly higher activity levels than control mice (*Tcf4^+/+^* and *Tcf4^+/+^::Neurod6-Cre*, *Figure 2—figure supplement 2A*) and similar activity levels as *Tcf4^STOP/+^* mice, indicating that embryonic reinstatement of *Tcf4* in forebrain glutamatergic neurons failed to rescue the hyperactivity phenotype (*Figure 2B*). Activity levels in *Tcf4^STOP/+^::Gad2-Cre* mice were statistically indistinguishable from either control mice (*Tcf4^+/+^* and *Tcf4^+/+^::Gad2-Cre*, *Figure 2—figure supplement 2B*) or *Tcf4^STOP/+^* mice (*Figure 2B*), suggesting that embryonic *Tcf4* reinstatement in GABAergic neurons is also insufficient to fully prevent the hyperactivity phenotype. *Tcf4* reinstatement in glutamatergic neurons improved object location memory, whereas *Tcf4* reinstatement in GABAergic neurons failed to fully prevent location memory impairments (*Figure 2C*). Importantly, lack of improvement in activity level and object location memory in *Tcf4^STOP/+^::Gad2-Cre* mice was reproduced by an independent investigator as part of the same study (*Figure 2—figure supplement 3A-B*). In the elevated plus maze task, we found that *Tcf4^STOP/+^::Neurod6-Cre* and control mice exhibited increased closed arm activity compared to *Tcf4^STOP/+^* mice, showing that reinstating *Tcf4* in glutamatergic neurons restored the low-anxiety phenotype (*Figure 2D*). In contrast, *Tcf4^STOP/+^::Gad2-Cre* and *Tcf4^STOP/+^* mice exhibited reduced closed arm activity compared to controls (*Figure 2D*), indicating persistence of the reduced anxiety-like phenotype despite reinstatement of *Tcf4* in GABAergic neurons. Finally, we observed that *Tcf4^STOP/+^::Neurod6-Cre* mice used a similar amount of nest materials as their respective controls, while *Tcf4^STOP/+^::Gad2-Cre* mice used more nest materials than *Tcf4^STOP/+^* mice, but significantly less material than controls (*Figure 2E*), demonstrating that embryonic reinstatement in glutamatergic neurons was sufficient to prevent the impaired nest building phenotype in PTHS model mice.

In addition to being expressed in neurons, *Tcf4* is expressed at all stages of oligodendrocyte development (*Kim et al., 2020*; *Phan et al., 2020*). Consistent with this expression pattern, the loss of TCF4 has been associated with decreased oligodendrocytes and impaired myelination (*Phan et al., 2020*). Therefore, one might expect that some behavioral phenotypes could be partially mitigated by reactivation of *Tcf4* expression in oligodendrocytes. To address this possibility, we re-expressed *Tcf4* in oligodendrocytes by crossing *Tcf4^STOP/+^* to *Olig2-Cre^+/-^* mice (*Tcf4^STOP/+^::Olig2-Cre*). We found that reinstating *Tcf4* in *Olig2*-expressing cells did not improve behavioral performance on open field and object location memory task in PTHS model mice (*Figure 2—figure supplement 3C*). In sum, our findings suggest that normalizing *Tcf4* expression from both glutamatergic and GABAergic neurons, and perhaps other cell types, might be required to fully rescue behavioral phenotypes.

## Neonatal intracerebroventricular administration of PHP.eB-hSyn-Cre produces widespread Cre expression in the brain during early postnatal development

We aimed to establish the extent to which postnatal reinstatement of *Tcf4* in neurons could prevent behavioral deficits in PTHS model mice. We conducted experiments designed to mimic an eventual viral-mediated gene therapy for PTHS in an idealistic manner with respect to reintroducing wildtype *Tcf4* isoforms and expression levels. To this end, we packaged a Cre transgene cassette into a recombinant AAV9-derived PHP.eB vector and bilaterally delivered this viral vector to the cerebral ventricles of neonates (*Figure 3A*). The Cre cassette was expressed under control of the human synapsin promoter (hSyn) for selective expression in neurons (*Nieuwenhuis et al., 2021*; *Figure 3—figure supplement 1*). We first examined Cre expression as a proxy for the temporal and spatial biodistribution of *Tcf4* reinstatement following intracerebroventricular (ICV) injection of a dose of $3.2 \times 10^{10}$ vector genome (vg) AAV9/PHP.eB-hSyn-Cre on postnatal day 1 (P1) (*Figure 3A*). We failed to detect significant Cre signals from relatively medial sagittal sections of P4 and P7 mouse brain, despite being able to observe a local distribution of Cre-expressing cells in the brain regions near the lateral ventricle injection site (*Figure 3B–C* and *Figure 3—figure supplement 1B-C*). Cre mRNA and protein distribution across the forebrain remained sparse until at least P10 (*Figure 3B–C* and *Figure 3—figure supplement 1D*). But, Cre was visibly and more broadly expressed in the hippocampus and throughout the cortical layers by P17 (*Figure 3B–C* and *Figure 3—figure supplement 1E*). The biodistribution of Cre protein at

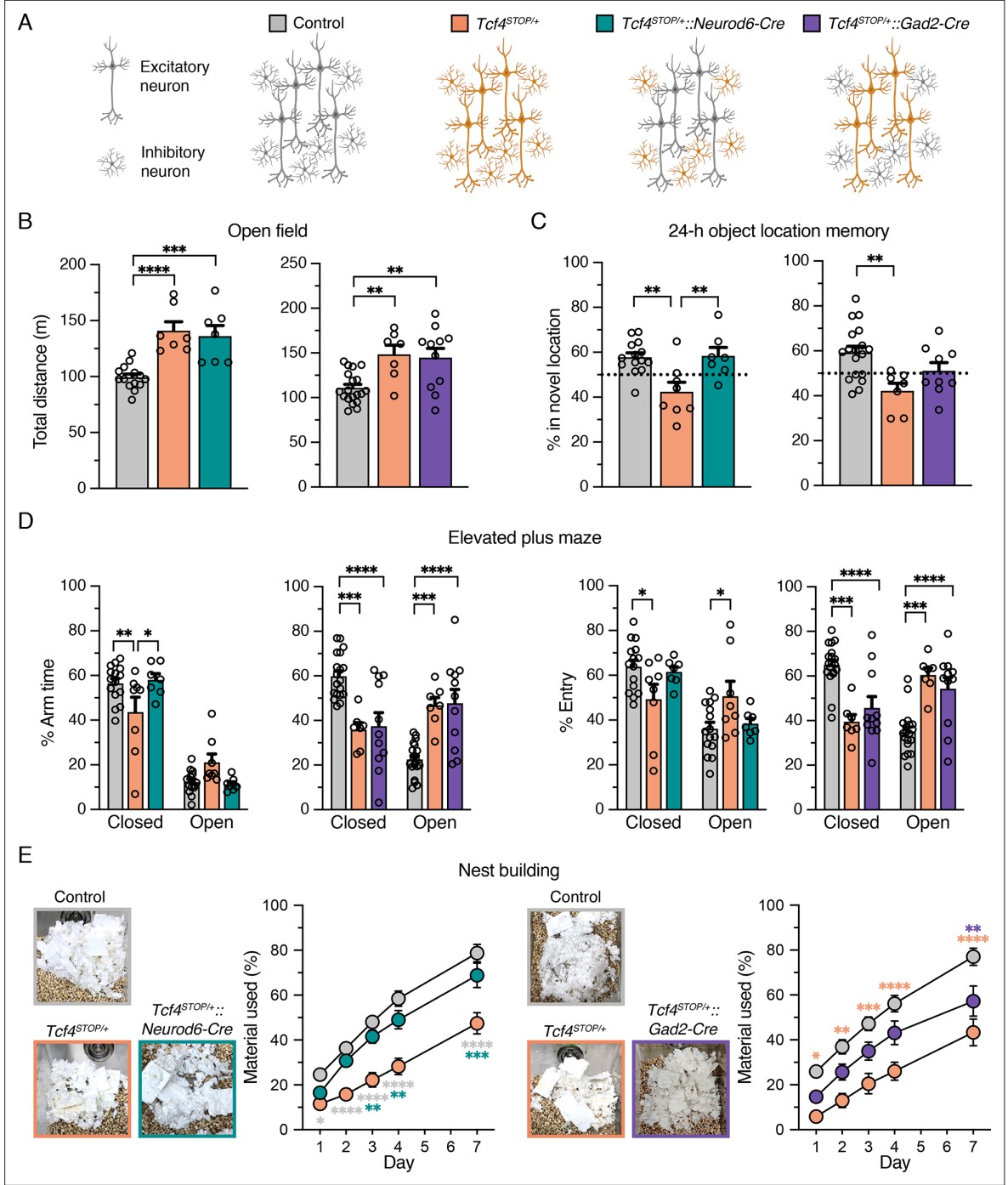

**Figure 2.** Embryonic reinstatement of *Tcf4* expression in glutamatergic or GABAergic neurons rescues selective behavioral deficits in a mouse model of PTHS. (**A**) Schematic representation of cell type-specific *Tcf4* reinstatement strategy. *Tcf4^STOP/+^::Neurod6-Cre* or *Tcf4^STOP/+^::Gad2-Cre* mice normalize *Tcf4* expression in glutamatergic or GABAergic neurons, respectively, while controls (*Tcf4^+/+^*, *Neurod6-Cre^+/–^*, or *Gad2-Cre^+/–^* mice) have normal *Tcf4* expression. (**B**) Total distance traveled for the 30 min testing period in the open field (Left panel: Control: n = 14, *Tcf4^STOP/+^*: n = 7, *Tcf4^STOP/+^::Neurod6-Cre*: n = 7, and right panel: Control: n = 19, *Tcf4^STOP/+^*: n = 7, *Tcf4^STOP/+^::Gad2-Cre*: n = 11). (**C**) Percent time interacting with the novel location object (Left panel: Control: n = 14, *Tcf4^STOP/+^*: n = 8, *Tcf4^STOP/+^::Neurod6-Cre*: n = 7, and right panel: Control: n = 18, *Tcf4^STOP/+^*: n = 7, *Tcf4^STOP/+^::Gad2-Cre*: n = 9). (**D**) Percent time spent in closed and open arms and percent entries made into the closed and open arms (Left panel: Control: n = 15, *Tcf4^STOP/+^*: n = 8, *Tcf4^STOP/+^::Neurod6-Cre*: n = 7, and right panel: Control: n = 19, *Tcf4^STOP/+^*: n = 7, *Tcf4^STOP/+^::Gad2-Cre*: n = 11). (**E**) Representative images of nests built by mice and percentage of nest material used during the 7 day nest building period (Left panel: Control: n = 15, *Tcf4^STOP/+^*: n = 8, *Tcf4^STOP/+^::Neurod6-Cre*: n = 7, and right panel: Control: n = 19, *Tcf4^STOP/+^*: n = 7, *Tcf4^STOP/+^::Gad2-Cre*: n = 11). Values are means ± SEM. *p < 0.05, **p < 0.005, ***p < 0.001, ****p < 0.0001.

*Figure 2 continued on next page*

*Figure 2 continued*

The online version of this article includes the following source data and figure supplement(s) for figure 2:

**Source data 1.** Numerical data shown in Figure 2.

**Figure supplement 1.** Single-cell RNA sequencing reveals cell type-specific *Tcf4* expression in the neonatal and adult mouse brain.

**Figure supplement 2.** Behavioral phenotypes are not affected in *Neurod6-Cre* [+/-] and *Gad2-Cre* [+/-] mice.

**Figure supplement 3.** Behavioral outcomes of *Tcf4*[STOP/+]*::Gad2-Cre* and *Tcf4*[STOP/+]*::Olig2-Cre* mice.

P60 was widespread in the brain, with particularly prominent expression in the forebrain compared to subcortical regions (*Figure 3D–F*), which is similar to patterns of endogenous TCF4 distribution (compare *Figure 3D* and *Figure 1—figure supplement 1A*).

To test whether Cre expression coincides with *Tcf4* reinstatement in *Tcf4*[STOP/+] mice, we quantified full-length *Tcf4* mRNA transcripts upon delivery of PHP.eB/Cre to *Tcf4*[STOP/+] neonates. The RT-qPCR results confirmed increased relative expression of *Tcf4* transcripts from P10 and P17 *Tcf4*[STOP/+] brains treated with PHP.eB/Cre (*Figure 3G*, *Tcf4*[STOP/+] + Vehicle at P10: 1.0 ± 0.09, n = 3; *Tcf4*[STOP/+] + PHP.eB/Cre at P10: 1.05 ± 0.05, n = 3; *Tcf4*[STOP/+] + Vehicle at P19: 1.0 ± 0.04, n = 4; *Tcf4*[STOP/+] + PHP.eB/Cre at P19: 1.39 ± 0.15, n = 3). Taken together, these observations confirm neonatal ICV injection to be a viable route of delivery to examine the behavioral consequences of postnatal *Tcf4* reinstatement in a subset of neurons.

## Postnatal reinstatement of *Tcf4* expression ameliorates behavioral phenotypes in PTHS model mice

We analyzed the behavioral performance of adult (P60 - P110) *Tcf4*[+/+] and *Tcf4*[STOP/+] mice after delivering vehicle or PHP.eB/Cre at P1 (*Figure 4A*). Similar to vehicle- or virally treated *Tcf4*[+/+] mice, *Tcf4*[STOP/+] mice treated with PHP.eB/Cre exhibited normal activity levels in the open field test, whereas vehicle-treated *Tcf4*[STOP/+] mice were hyperactive (*Figure 4B*). In addition to normalizing activity levels, PHP.eB/Cre treatment also fully rescued long-term memory performance in *Tcf4*[STOP/+] mice compared to vehicle-treated *Tcf4*[STOP/+] mice (*Figure 4C*). In the elevated plus maze, PHP.eB/Cre-treated *Tcf4*[STOP/+] mice spent relatively more time in the closed arms than the open arms, similar to vehicle- and PHP.eB/Cre-treated *Tcf4*[+/+] mice. In contrast, vehicle-treated *Tcf4*[STOP/+] mice spent similar time in the open and closed arms as a sign of their abnormally low anxiety levels (*Figure 4D*). Lastly, we found that PHP.eB/Cre treatment supported progressive improvement of nest building behavior in *Tcf4*[STOP/+] mice. Specifically, although both *Tcf4*[STOP/+] groups appeared to be similarly lacking in the execution of this behavior at baseline, but over the course of 1 week, PHP.eB/Cre-treated *Tcf4*[STOP/+] mice proved capable of incorporating nest materials to a similar degree as vehicle- or virally treated *Tcf4*[+/+] mice (*Figure 4E*).

While PHP.eB/Cre-mediated postnatal reinstatement of *Tcf4* partially or fully recovered performance on a variety of behavioral phenotypes in PTHS model mice, we found that small body and brain sizes were not corrected (*Figure 4F–G*). This suggests that earlier intervention, improved pan-cellular biodistribution, and/or *Tcf4* reinstatement in other cell types such as oligodendrocytes might be necessary to restore anatomical integrity, although this is evidently not a prerequisite for recovery of behavioral phenotypes.

Our data collectively demonstrate the potential for virally mediated postnatal normalization of *Tcf4* expression to broadly rescue behavioral phenotypes. However, to exclude the possibility that viral infection itself might rescue behavioral phenotypes in PTHS model mice, we analyzed behavioral phenotypes from *Tcf4*[+/+] and *Tcf4*[STOP/+] mice after delivering AAV9/PHP.eB-hSyn-Green Fluorescence Protein (GFP) at P1. Treating *Tcf4*[STOP/+] group with PHP.eB/GFP did not rescue abnormal behavioral phenotypes (*Figure 4—figure supplement 1*), indicating that behavioral rescue in PTHS model mice was driven by Cre-mediated excision of the STOP cassette in the *Tcf4* allele.

Our data show that targeting a postnatal reinstatement of *Tcf4* broadly in neuronal cell types can provide therapeutic benefit. Nonetheless, we additionally tested whether a postnatal reinstatement of *Tcf4* selectively in forebrain excitatory neurons might reverse behavioral phenotypes. To accomplish this, we crossed our conditional mice to *Camk2a-Cre*[+/-] transgenic mice to reactivate *Tcf4* expression selectively in *Camk2a*-expressing neurons during postnatal development (*Tcf4*[STOP/+]*::Camk2a-Cre*) (*Tsien et al., 1996*). We found that long-term memory deficit and abnormal innate behavior, but not the hyperactive phenotype, were rescued in *Tcf4*[STOP/+]*::Camk2a-Cre* mice (*Figure 4—figure*

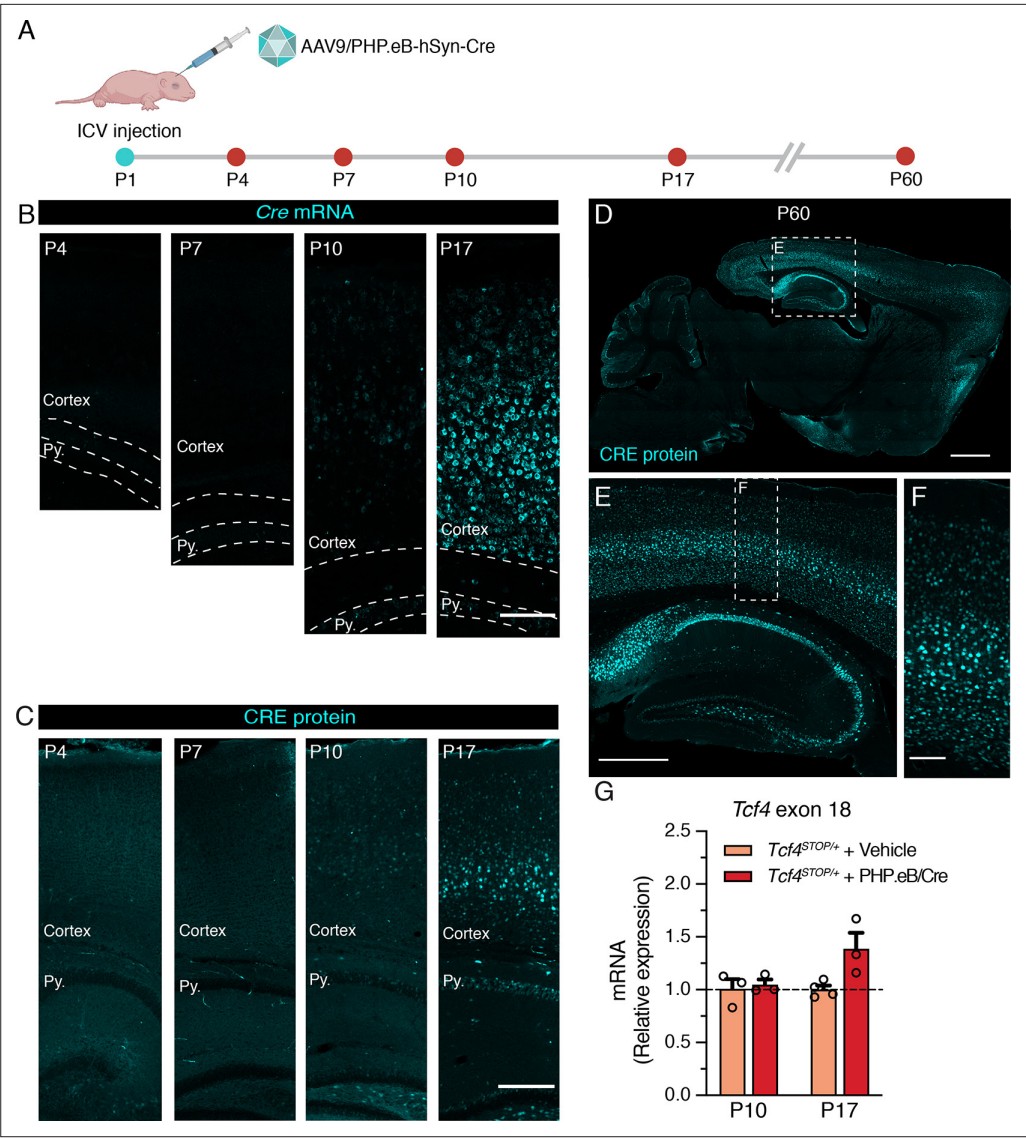

**Figure 3.** Neonatal ICV delivery of PHP.eB/Cre yields Cre expression by approximately P10-P17. (**A**) A timeline of experiment to evaluate timing of Cre biodistribution following intracerebroventricular (ICV) injection of 1 µl of 3.2 × 10¹³ vg/ml AAV9/PHP.eB-hSyn-Cre to P1 mice. (**B**) In situ hybridization for *Cre* mRNA, and (**C**) immunofluorescence staining for CRE protein in the cortex and hippocampus of P4, P7, P10, and P17 wildtype mice neonatally treated with PHP.eB/Cre. Py. = Stratum pyramidale. Scale bars = 100 µm (**B**) and 250 µm (**C**). (**D–F**) CRE immunofluorescence staining in sagittal section of P60 wildtype mouse brain. Scale bars = 1 mm (**D**), 500 µm (**E**), and 100 µm (**F**). (**G**) Relative *Tcf4* transcript levels detected in the brains of P10 *Tcf4^STOP/+^* mice treated with vehicle or PHP.eB/Cre and P17 *Tcf4^STOP/+^* mice treated with vehicle or PHP.eB/Cre.

The online version of this article includes the following source data and figure supplement(s) for figure 3:

**Source data 1.** Numerical data shown in *Figure 3*.

**Figure supplement 1.** Cre immunofluorescence staining in sagittal sections of P4, P7, P10, and P17 mice.

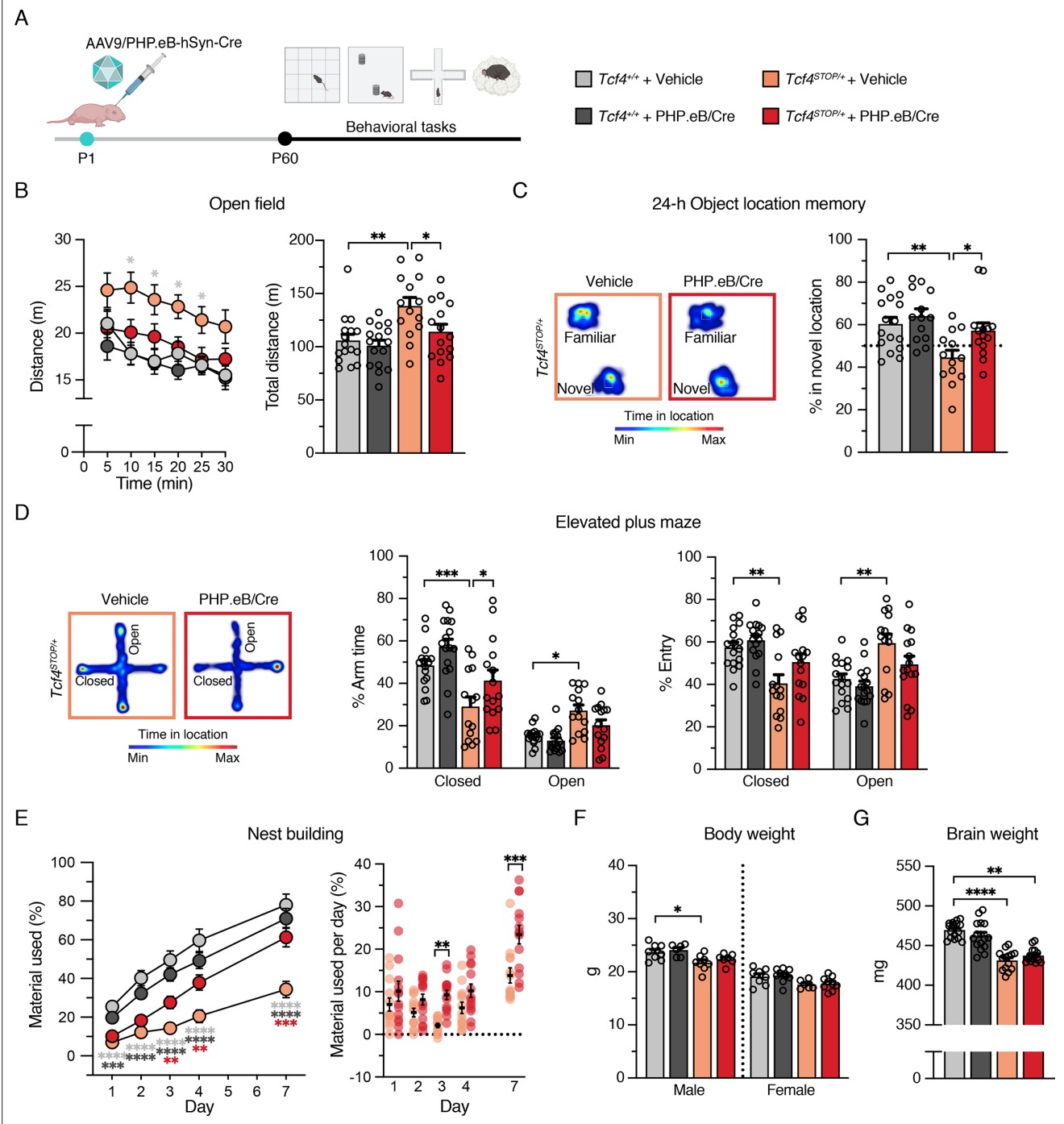

**Figure 4.** Neonatal ICV injection of PHP.eB/Cre improves behavioral phenotypes in *Tcf4STOP/+* mice. (**A**) Experimental timeline for evaluation of behavioral phenotypes in *Tcf4+/+* and *Tcf4STOP/+* mice treated with vehicle or PHP.eB/Cre. (**B**) Left panel: Distance traveled per 5 min. Right panel: Total distance traveled for the 30-min testing period. (**C**) Left panel: Heatmaps indicate time spent in proximity to one object located in the familiar position and the other object relocated to a novel position. Right panel: Percent time interacting with the novel location object. (**D**) Left panel: Heatmaps reveal time spent in elevated plus maze. Right panels: Percent time spent in the closed and open arms and percent entries made into the closed and open arms. (**E**) Left panel: Percentage of nest material used during the 7-day nest building period. Right panel: Percentage of nest material used per day. (**F**) Body weight analysis of P65-69 male and female mice. (**G**) Adult brain weight analysis. Values are means ± SEM. *p < 0.05, **p < 0.005, ***p < 0.001, ****p < 0.0001.

The online version of this article includes the following source data and figure supplement(s) for figure 4:

**Source data 1.** Mean and SD data of each biological replicate for *Figure 4*.

*Figure 4 continued on next page*

*Figure 4 continued*

**Source data 2.** Numerical data shown in *Figure 4*.

**Figure supplement 1.** Behavioral outcomes of *Tcf4*$^{+/+}$ and *Tcf4*$^{STOP/+}$ treated with PHP.eB/GFP.

**Figure supplement 2.** Behavioral outcomes of *Tcf4*$^{STOP/+}$*::Camk2a-Cre* mice.

*supplement 2*). These findings reinforce the idea that targeting *Tcf4* reinstatement in both excitatory and inhibitory neurons, rather than one of these neuronal classes, provides better therapeutic outcomes for a PTHS genetic therapy.

## Postnatal *Tcf4* reinstatement partially corrects local field potential abnormalities in PTHS model mice

Several clinical observations have reported electroencephalographic (EEG) abnormalities, such as altered slow waves, in individuals with PTHS (*Amiel et al., 2007*; *Peippo et al., 2006*; *Takano et al., 2010*), yet these phenotypes have not been examined in PTHS model mice. Here we performed local field potential (LFP) recordings in *Tcf4*$^{STOP/+}$ mice, which provide an accurate indication of local neuronal activity (*Buzsáki et al., 2012*). We implanted recording electrodes in the hippocampus, a site of high *Tcf4* expression (*Kim et al., 2020*) and a region in which there is a well-characterized enhancement of long-term potentiation in PTHS model mice (*Kennedy et al., 2016*; *Thaxton et al., 2018*), and then recorded LFPs from freely moving mice (*Figure 5A* and *Figure 5—figure supplement 1A-B*). We observed a trend for reduced total LFP power in *Tcf4*$^{STOP/+}$ mice, but the total power between *Tcf4*$^{+/+}$ and *Tcf4*$^{STOP/+}$ mice was not statistically distinguishable (*Figure 5—figure supplement 1C-D*). Significant decreases in *Tcf4*$^{STOP/+}$ LFP power were evident in the theta (5–8 Hz) band (*Figure 5—figure supplement 1C, E*). A moderate but consistent decrease in power likely underlies this phenotype as per follow-up spectrogram analyses (*Figure 5—figure supplement 1F*). Having established that *Tcf4* haploinsufficiency resulted in LFP abnormalities in mice, we sought to determine whether LFP power could be normalized by postnatal *Tcf4* reinstatement. Upon analyzing LFP recordings in vehicle- and virus-treated groups (*Figure 5B–C*), we found total LFP power to be significantly reduced in vehicle-treated *Tcf4*$^{STOP/+}$ mice compared to vehicle- and virally-treated *Tcf4*$^{+/+}$ mice, but partially normalized in PHP.eB/Cre-treated *Tcf4*$^{STOP/+}$ mice (*Figure 5D*). This effect appeared to be largely driven by the normalization of theta band activity (*Figure 5E*), which was evident across one-minute recording epochs (*Figure 5F*). Collectively, these data define hippocampal LFP deficits in PTHS model mice and demonstrate their amenability to normalization by postnatal reinstatement of *Tcf4*.

## Postmortem evaluation of *Cre* biodistribution and expression of *Tcf4* and TCF4-regulated genes

After completing behavioral and LFP experiments, we performed in situ hybridization (ISH) to characterize *Cre* distribution and RT-qPCR to examine effectiveness of PHP.eB/Cre treatment on expression levels of *Tcf4* and TCF4-regulated genes. ISH revealed that *Cre* mRNA was still robustly detected in 6-month-old mice, subsequent to viral delivery of the *Cre* transgene at P1 (*Figure 6A–D*). In the cortex and olfactory bulb, *Cre* was observed in most cells, but certainly not all, throughout the layers (*Figure 6C–D*). Similarly, although we found most pyramidal cells to express *Cre* in the hippocampus, non-expressing cells were evident, especially in the dentate gyrus (*Figure 6B*). This observation suggests that reinstating *Tcf4* in a subset of neurons can provide therapeutic benefit. To ensure the relative uniformity of *Cre* transduction among treated *Tcf4*$^{STOP/+}$ mice, we analyzed *Cre* fluorescence in the neocortex and pyramidal cell layer of CA1. Levels proved generally consistent, with the exception of one mouse whose neocortex and CA1 *Cre* fluorescence was ~3–4 times higher than the group median (*Figure 6E*).

We next analyzed *Tcf4* mRNA expression from the forebrains of the PHP.eB/Cre-treated *Tcf4*$^{STOP/+}$ mice. On average, *Tcf4* mRNA expression was approximately 1.3-fold higher in virally treated versus vehicle-treated *Tcf4*$^{STOP/+}$ forebrains at P150-P200 (*Figure 6F*, *Tcf4*$^{+/+}$ + Vehicle: $1.0 \pm 0.18$, n = 15; *Tcf4*$^{STOP/+}$ + Vehicle: $0.56 \pm 0.09$, n = 14; *Tcf4*$^{STOP/+}$ + PHP.eB/Cre: $0.72 \pm 0.12$, n = 15). Recent RNA-sequencing studies revealed genes whose expression levels were altered by heterozygous *Tcf4* disruption (*Kennedy et al., 2016*; *Phan et al., 2020*). To analyze the impact of PHP.eB/Cre treatment on the expression of TCF4-regulated genes, we examined expression levels of the following genes:

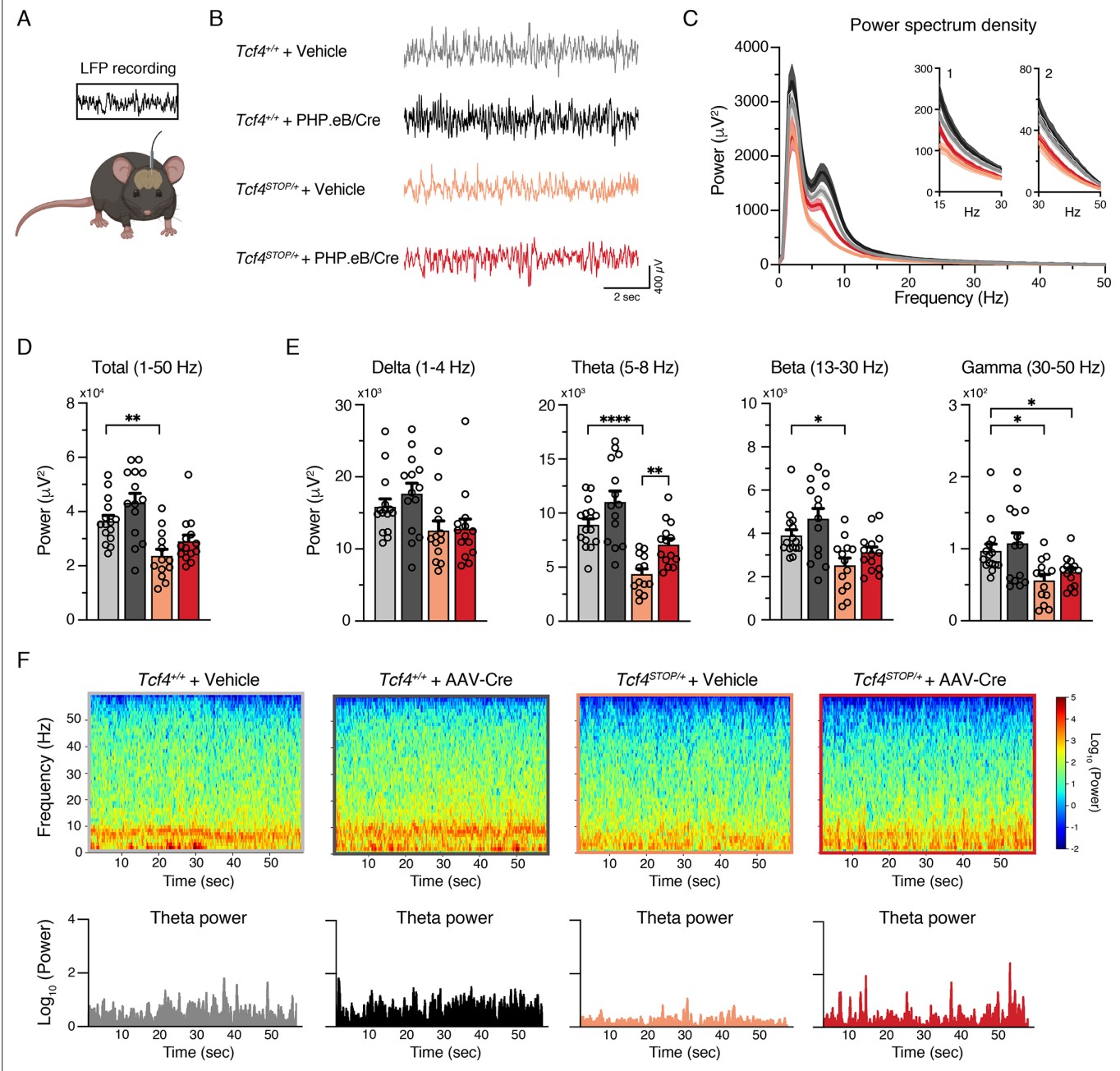

**Figure 5.** Neonatal ICV injection of PHP.eB/Cre partially rescues LFP spectral power in *Tcf4^STOP/+* mice. (**A**) Schematic of local field potential (LFP) recording from the hippocampus of a freely moving mouse. (**B**) Representative examples of LFP in each experimental group. (**C**) Power spectrum density of hippocampal LFP analyzed from *Tcf4^+/+* and *Tcf4^STOP/+* mice treated with vehicle or PHP.eB/Cre. Inset 1 spans from 15 to 30 Hz. Inset 2 spans from 30 to 50 Hz on x-axis. (**D**) LFP power analyses of frequency bands ranging from 1 to 50 Hz, (**E**) delta (1–4 Hz), theta (5–8 Hz), beta (13–30 Hz), and gamma (30–50 Hz) bands (*Tcf4^+/+* + vehicle: n = 15, *Tcf4^+/+* + PHP.eB/Cre: n = 14, *Tcf4^STOP/+* + vehicle: n = 13, and *Tcf4^STOP/+* + PHP.eB/Cre: n = 14). (**F**) Top panels: Spectrograms in single LFP sessions of representative experimental groups. Bottom panels: Representative theta power extracted from spectrogram in the top panel. Values are means ± SEM. *p < 0.05, **p < 0.005, ****p < 0.0001.

The online version of this article includes the following source data and figure supplement(s) for figure 5:

**Source data 1.** Numerical data shown in *Figure 5*.

**Figure supplement 1.** *Tcf4* haploinsufficiency alters LFP spectral power in the theta band.

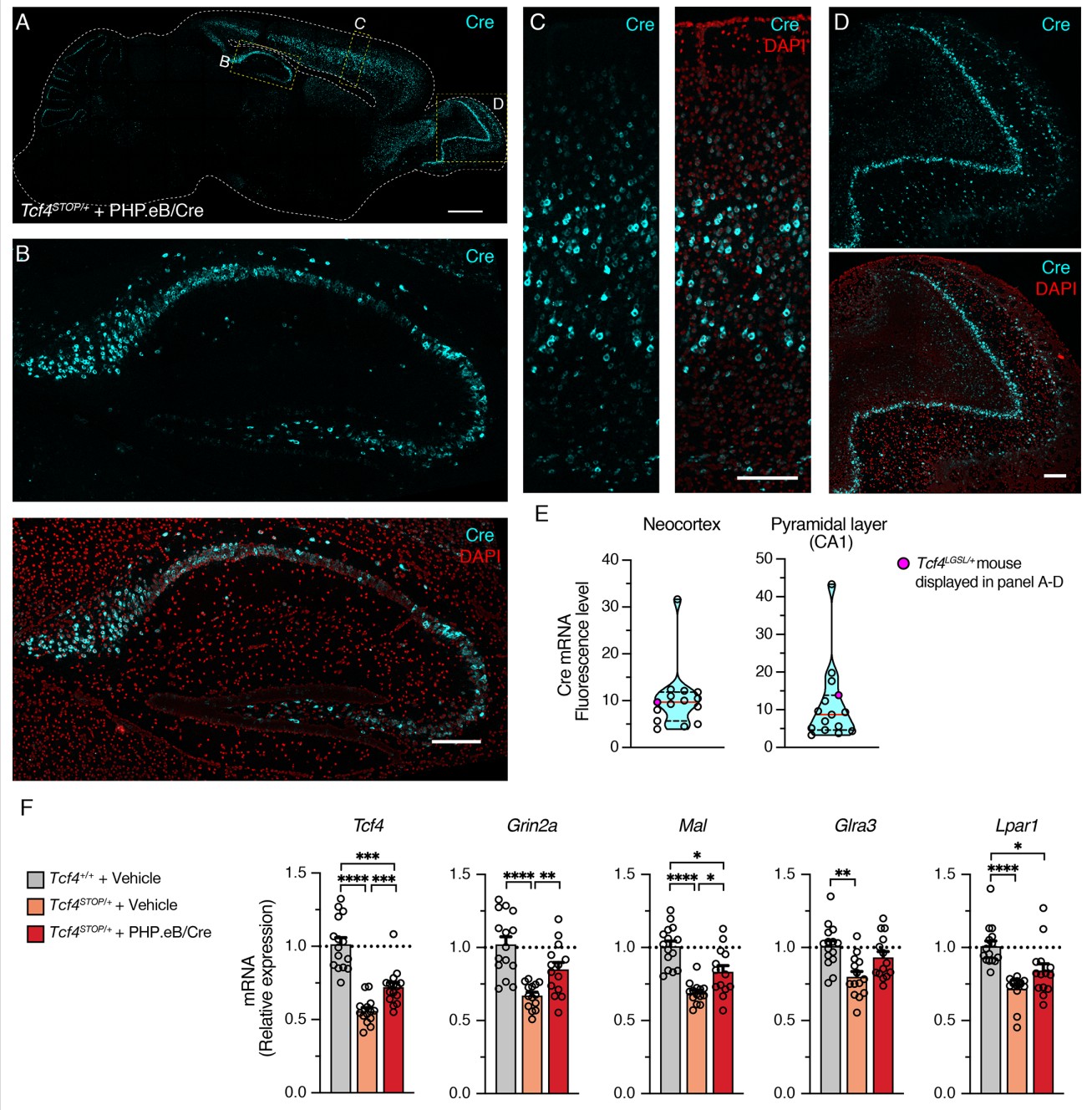

**Figure 6.** Widespread *Cre* expression of the forebrain leads to partial upregulation of *Tcf4* and partial recovery of selected TCF4-regulated gene expression. (**A**) Representative image of in situ hybridization for *Cre* mRNA in sagittal section of 6-month-old *Tcf4^{STOP/+}* mouse that was treated at P1 with PHP.eB/Cre. Scale bar = 1 mm. (**B–D**) Higher magnification images of boxed regions in panel A. Scale bars = 200 μm. (**E**) *Cre* mRNA fluorescence levels of neocortex and CA1 pyramidal cell layer analyzed from individual *Tcf4^{STOP/+}* + PHP.eB/Cre mice. The red line and black dotted lines of the violin plot represent median and interquartile ranges of the data, respectively. (**F**) Relative *Tcf4* mRNA expression of the forebrain from vehicle-treated *Tcf4^{+/+}* (n = 15), vehicle-treated *Tcf4^{STOP/+}* (n = 14), and PHP.eB/Cre-treated *Tcf4^{STOP/+}* (n = 15) mice. *Tcf4* mRNA expression levels of PHP.eB/Cre-treated *Tcf4^{STOP/+}* mice are relatively higher than vehicle-treated *Tcf4^{STOP/+}* mice. Relative mRNA expressions of selected TCF4-regulated genes: *Grin2a* (*Tcf4^{+/+}* + Vehicle: 1.0 ± 0.05, n = 15; *Tcf4^{STOP/+}* + Vehicle: 0.67 ± 0.02, n = 14; *Tcf4^{STOP/+}* + PHP.eB/Cre: 0.85 ± 0.05, n = 14), *Mal* (*Tcf4^{+/+}* + Vehicle: 1.0 ± 0.04, n = 15; *Tcf4^{STOP/+}* + Vehicle: 0.70 ± 0.02, n = 14; *Tcf4^{STOP/+}* + PHP.eB/Cre: 0.83 ± 0.04, n = 14), *Glra3* (*Tcf4^{+/+}* + Vehicle: 1.0 ± 0.04, n = 15; *Tcf4^{STOP/+}* + Vehicle: 0.80 ± 0.04, n = 14; *Tcf4^{STOP/+}* + PHP.eB/Cre: 0.93 ± 0.04, n = 14), and *Lpar1* (*Tcf4^{+/+}* + Vehicle: 1.0 ± 0.04, n = 15; *Tcf4^{STOP/+}* + Vehicle: 0.71 ± 0.03, n = 14; *Tcf4^{STOP/+}* + PHP.eB/Cre: 0.84 ± 0.05, n = 14). Values are means ± SEM. *p < 0.05, **p < 0.005, ***p < 0.001, ****p < 0.0001.

The online version of this article includes the following source data for figure 6:

**Source data 1.** Numerical data shown in *Figure 6*.

*Grin2a* (encoding for NMDA receptor subunit epsilon-1), *Mal* (encoding for myelin and lymphocyte protein), *Glra3* (encoding for glycine receptor subunit alpha-3), and *Lpar1* (encoding for lysophosphatidic acid receptor 1), all genes whose expression has been shown to be dysregulated by *Tcf4* haploinsufficiency (**Kennedy et al., 2016**; **Phan et al., 2020**). Each was noticeably downregulated in vehicle-treated *Tcf4*$^{STOP/+}$ mice, but at least partially normalized by PHP.eB/Cre treatment (**Figure 6F**). Our postmortem analyses demonstrated partially normalized expression levels of both *Tcf4* and TCF4-targeted genes, which appear to be driven by a subset of neurons whose *Tcf4* expression was fully normalized to its wildtype levels. These observations suggest that normalizing *Tcf4* expression in a subset of neurons might be sufficient to rescue behavioral phenotypes in *Tcf4*$^{STOP/+}$ mice.

## Discussion

The genetic mechanism of PTHS suggests a therapeutic opportunity; loss-of-function in one *TCF4* copy is sufficient to cause PTHS, so conversely, restoring TCF4 function could treat PTHS. In this proof-of-concept study, genetic reinstatement of *Tcf4* in a subset of neurons during early postnatal development corrected multiple behavioral phenotypes in PTHS model mice, including hyperactivity, reduced anxiety-like behavior, memory deficit, and abnormal innate behavior. Furthermore, early postnatal *Tcf4* reinstatement corrected altered local field potential activity and, at the molecular level, TCF4-regulated gene expression changes. Our results suggest that postnatal genetic therapies to compensate for loss-of-function of TCF4 can offer an effective treatment for PTHS.

One of the key parameters for genetic normalization strategies is the age at time of intervention. We likely accomplished moderately widespread *Tcf4* normalization throughout the brain by P10-P17 (**Figure 3**), which is roughly equivalent to the first 2 years of human life (**Wang et al., 2020**). Individuals with PTHS have global developmental delay, often presenting itself in the first year of life (**Zollino et al., 2019**). Thus, our study indicates that genetic normalization approaches could provide a viable early life treatment opportunity. Future investigation is needed to address the extent to which juvenile and adult *Tcf4* reinstatement can correct behavioral phenotypes, as this could provide clinical insights into whether later-onset interventions might also provide therapeutic benefits.

While it is important to define the developmental therapeutic window, eventual ASO- or AAV-mediated genetic therapies will also need to produce an appropriate biodistribution to be efficacious. Previous studies have shown that *Tcf4* expression levels are particularly high in the forebrain (**Kim et al., 2020**). Accordingly, our proof-of-concept study employed a strategy to reinstate *Tcf4* expression more prominently in the forebrain than in subcortical regions (**Figures 3 and 6**). Notably, intrathecal delivery of ASOs produces a biodistribution favoring the forebrain (**Mazur et al., 2019**), remarkably similar to the endogenous *Tcf4* expression pattern (**Kim et al., 2020**). Given our ability to recover behavioral phenotypes, future efforts to test the feasibility of ASOs or gene therapies should target *Tcf4* reinstatement in the forebrain to provide therapeutic benefit.

Another important consideration for eventual genetic therapies is which cell types should be targeted to rescue behavioral phenotypes. The therapeutic potential of genetic rescue in specific cell types had not been evaluated due to the lack of conditional models for *Tcf4* reinstatement. Our conditional restoration model provides a powerful tool in that we can establish the cellular and behavioral impacts of cell-type-specific *Tcf4* restoration, which will ultimately inform therapeutic development for PTHS. Because *Tcf4* is expressed at particularly high levels in glutamatergic and GABAergic neurons during embryonic development (**Jung et al., 2018**) and throughout the postnatal period (**Kim et al., 2020**; **Figure 2—figure supplement 1**), we initially aimed to embryonically reinstate TCF4 function in these broad neuronal classes to establish their relative contribution to behavioral rescue. Reactivating *Tcf4* expression in excitatory pyramidal neurons, dentate gyrus mossy cells, and granule cells within the dorsal telencephalon, starting from ~E11.5, improved memory, anxiety phenotype, and innate behavior, while reactivating *Tcf4* expression in almost all GABAergic neurons throughout the brain at ~E13.5 partially rescued memory and innate behavior (**Goebbels et al., 2006**; **Taniguchi et al., 2011**; **Figure 2**). Notably, the hyperactivity phenotype was not rescued by reinstating *Tcf4* expression from either of the neuronal classes alone, suggesting that hyperactivity phenotype could be dependent on both excitatory and inhibitory neurons having a normal complement of *Tcf4* to function properly. Given that memory, anxiety-like behavior, and innate behavior were fully rescued when *Tcf4* was re-activated only from excitatory neurons, it is reasonable to argue that these phenotypes might mainly depend on having normal TCF4 levels in excitatory neurons.

In order to provide a complete understanding of the interplay between excitatory and inhibitory neurons in behavioral outputs, the effect of cell-type-specific deletion of *Tcf4* on behavioral phenotypes needs to be further evaluated.

Due to the established role for TCF4 in regulating the maturation of oligodendrocyte progenitors (*Phan et al., 2020*), it is tempting to speculate that normalizing *Tcf4* expression in oligodendrocytes might be important for recovery of behavioral phenotypes. However, when *Tcf4* was reactivated in oligodendrocytes, oligodendrocyte precursor cells, and a subpopulation of astrocytes (*Wang et al., 2021*), hyperactivity and spatial memory deficits were not recovered (*Figure 2—figure supplement 3*). Therefore, our findings emphasize that TCF4 loss in neurons contributes to PTHS-associated behavioral phenotypes, and therapeutic strategies should thus target the normalization of *Tcf4* expression in neurons. Accordingly, we demonstrated, at the proof-of-concept level, that mosaic, neuronal reinstatement of normal TCF4 levels can produce phenotypic rescue (*Figures 4–5*). However, it is possible that greater phenotypic rescue could have been achieved with additional reinstatement of *Tcf4* in oligodendrocytes and perhaps other cell classes. Finally, it is important to recognize that TCF4 protein is found outside of the brain (*Sepp et al., 2011*). Thus, it is plausible that increasing TCF4 levels in other regions of the body might provide therapeutic benefit, particularly for phenotypes such as constipation that might have a peripheral contribution.

A *TCF4* normalization treatment strategy has an advantage in that it addresses the core genetic defect in PTHS, and therefore should restore transcriptional targets of TCF4. *Tcf4* haploinsufficiency in mice alters the expression of genes that are involved in synaptic plasticity and neuronal excitability, such as *Grin2a* and *Glra3*, and neuronal development, such as *Mal* and *Lpar1* (*Kennedy et al., 2016*; *Phan et al., 2020*). Our data show that upregulating *Tcf4* levels postnatally can correct the expression of these TCF4-regulated genes in the brain (*Figure 6*). The effect of *Tcf4* normalization on downstream genes might help guide future preclinical studies. For example, therapeutic agent choice and their dosing could be optimized by testing behavioral recovery and by measuring expression of TCF4-regulated genes, such as those validated in this study.

Our study has several important limitations. First, we normalized *Tcf4* expression only in neurons. *Tcf4* is expressed in nearly all neurons, astrocytes, and oligodendrocytes (*Jung et al., 2018*; *Kim et al., 2020*). Ideally, *Tcf4* should be reinstated in both neuronal and non-neuronal cells to accomplish maximum therapeutic outcomes. Our preliminary data indicated that injecting AAV containing a broadly active promoter, CAG, reinstated *Tcf4* in all cell types, but induced abnormal glial activation, which is reminiscent of toxicity previously reported with CAG vectors (*Xiong et al., 2019*), and severe weight loss (data not shown). To avoid toxicity in our experimental paradigm while still achieving efficient transduction, we employed the neuron-selective promoter (hSyn) in our viral construct. Nonetheless, our data provide compelling evidence that reinstating *Tcf4* only in neurons is sufficient to reverse behavioral and LFP phenotypes in PTHS model mice. Second, our study does not inform therapeutic threshold that must be achieved by genetic normalization approaches. TCF4 is a dosage-sensitive protein: too little expression causes neurodevelopmental disorders, and too much expression may be linked to schizophrenia (*Brennand et al., 2011*; *Brzózka et al., 2010*; *Forrest et al., 2012*; *Quednow et al., 2014*; *Sepp et al., 2012*; *Wirgenes et al., 2012*). Our conditional model allowed us to establish the best-case treatment scenario by reinstating *Tcf4* to wild-type levels. Future proof-of-concept preclinical studies to upregulate *Tcf4* through ASOs or gene therapy approaches in PTHS model mice must take considerable care to recapitulate optimal levels of *Tcf4* expression. Third, our study does not provide insights into which isoforms are most appropriate to deliver to the brain through AAV-mediated gene therapy. The *TCF4* gene produces at least 18 isoforms, which may have cell type- and developmental-specific expression patterns in the brain (*Sepp et al., 2011*). Characterizing endogenous isoform expression in the human brain will be critical to guide design of a viral vector that produces appropriate *TCF4* isoform expression.

To our knowledge, the present study is the first investigation to show that normalizing *Tcf4* expression during early postnatal development can improve behavioral outcomes in a mouse model of PTHS. Furthermore, our studies provide insights into target cell types and biodistribution that must be achieved for therapeutic recovery in mice, which guides the rational design of genetic normalization approaches such as AAV-mediated gene therapy, ASOs, or small molecules. In sum, our findings suggest parameters for which genetic therapies can provide substantial therapeutic benefit for individuals with PTHS.

# Materials and methods

## Key resources table

| Reagent type (species) or resource | Designation | Source or reference | Identifiers | Additional information |
|---|---|---|---|---|
| Gene (*M. musculus*) | Transcription factor 4 (Tcf4) | GenBank | MGI:MGI:98,506 | |
| Genetic reagent (*M. musculus*) | (C57BL/6 J) *Tcf4*$^{STOP/+}$ | doi:10.3389/fnana2020.00042 | | |
| Genetic reagent (*M. musculus*) | (C57BL/6 J) *Actb-Cre*$^{+/-}$ | Jackson Laboratory | Strain #: 019099 | PMID:9598348 |
| Genetic reagent (*M. musculus*) | (C57BL/6 J) *Gad2-Cre*$^{+/-}$ | Jackson Laboratory | Strain #: 010802 | PMID:21943598 |
| Genetic reagent (*M. musculus*) | (C57BL/6 J) *Olig2-Cre*$^{+/-}$ | Jackson Laboratory | Strain #: 025567 | PMID:20569695 |
| Genetic reagent (*M. musculus*) | (C57BL/6 J) *Camk2a-Cre*$^{+/-}$ | Jackson Laboratory | Strain #: 005359 | PMID:8980237 |
| Genetic reagent (*M. musculus*) | (C57BL/6 J) *Neurod6-Cre*$^{+/-}$ | doi:10.1002/dvg.20256 | | |
| Antibody | Mouse monoclonal anti-Cre recombinase | Millipore Sigma | MAB3120 | (1:1000) |
| Antibody | Guinea pig polyclonal anti-NeuN | Millipore Sigma | ABN90P | (1:1000) |
| Antibody | Rabbit polyclonal anti-GFAP | Agilent Dako | Z033429-2 | (1:1000) |
| Antibody | Rabbit polyclonal anti-GFP | NOVUS Biologicals | NB600-308 | (1:1000) |
| Antibody | Goat polyclonal anti-rabbit Alexa 568 | Invitrogen | A11011 | (1:1000) |
| Antibody | Goat polyclonal anti-mouse Alexa 647 | Invitrogen | A21240 | (1:1000) |
| Antibody | Goat polyclonal anti-guinea pig Alexa 594 | Invitrogen | A11076 | (1:1000) |
| Antibody | Goat polyclonal anti-rabbit Alexa 448 | Invitrogen | A32731 | (1:1000) |
| Other | Biotinylated goat anti-rabbit antibody | Vector Laboratories | BA-1000–1.5 | Biotinylated secondary antibody, (1:500) |
| Other | ABC elite avidin-biotin-peroxidase system | Vector Laboratories | Vector PK-7100 | Detection of biotinylated molecule |
| Other | DAPI stain | Invitrogen | D1306 | Blue-fluorescent DNA stain (700 ng/m) |
| Antibody | Mouse monoclonal anti-ITF-2 | Santa Cruz Biotechnology | Sc-393407 | (1:1000), doi: 10.1523/ENEURO.0197–21.2021 |
| Antibody | Rabbit polyclonal anti-beta Tubulin | Abcam | Ab6046 | (1:5000) |
| Antibody | Goat polyclonal anti-mouse secondary antibody, HRP | Thermo Fisher | 31,430 | (1:5000) |
| Antibody | Goat polyclonal anti-rabbit secondary antibody, HRP | Thermo Fisher | 31,460 | (1:5000) |
| Other | Bicinchoninic acid assay | Thermo Scientific | 23,225 | Quantitation of total protein |
| Other | Protease inhibitor cocktail | Sigma | P8340 | Inhibition of serine-proteases |
| Other | Odyssey blocking buffer | Li-COR Biosciences | 927–40100 | Phosphate-buffered saline that provides optimal blocking conditions |

*Continued on next page*

*Continued*

| Reagent type (species) or resource | Designation | Source or reference | Identifiers | Additional information |
|---|---|---|---|---|
| Other | Polyvinylidene fluoride membranes | Fisher Scientific | 45-004-110 | Membrane materials used for Western blot |
| Commercial assay, kit | Clarity Western ECL Substrate | Bio-Rad | 1705061 | |
| Commercial assay, kit | RNAscope Fluorescent Multiplex Assay | Advanced Cell Diagnostics | 320,850 | |
| Commercial assay, kit | RNAscope Protease IV | Advanced Cell Diagnostics | 322,340 | |
| Commercial assay, kit | RNAscope Probe-iCRE-C3 | Advanced Cell Diagnostics | 423321-C3 | Accession No:AY056050.1 |
| Commercial assay, kit | RNeasy Mini Kit | Qiagen | 74,106 | |
| Commercial assay, kit | SYBR green master mix | Thermofisher | A25742 | |
| Other | cDNA SuperMix | QuantaBio | 101414–106 | First-strand cDNA synthesis |
| Sequence-based reagent | mTcf4 Forward | This paper | PCR primers | GGGAGGAAGAGAAGGTGT |
| Sequence-based reagent | mTcf4 Reverse | This paper | PCR primers | CATCTGTCCCATGTGATTCGC |
| Sequence-based reagent | *Grin2a* Forward | This paper | PCR primers | TTCATGATCCAGGAGGAGTTTG |
| Sequence-based reagent | *Grin2a* Reverse | This paper | PCR primers | AATCGGAAAGGCGGAGAATAG |
| Sequence-based reagent | *Mal* Forward | This paper | PCR primers | CTGGCCACCATCTCAATGT |
| Sequence-based reagent | *Mal* Reverse | This paper | PCR primers | TGGACCACGTAGATCAGAGT |
| Sequence-based reagent | *Glra3* Forward | This paper | PCR primers | GGGCATCACCACTGTACTTA |
| Sequence-based reagent | *Glra3* Reverse | This paper | PCR primers | CCGCCATCCAAATGTCAATAG |
| Sequence-based reagent | *Npar1* Forward | This paper | PCR primers | CCCTCTACAGTGACTCCTACTT |
| Sequence-based reagent | *Npar1* Reverse | This paper | PCR primers | GCCAAAGATGTGAGCGTAGA |
| Sequence-based reagent | *Actin* Forward | This paper | PCR primers | GGCACCACACCTTCTACAATG |
| Sequence-based reagent | *Actin* Reverse | This paper | PCR primers | GGGGTGTTGAAGGTCTCAAAC |
| Software, algorithm | Ethovision XT 15.0 | Noldus | | |
| Software, algorithm | Spike2 | Cambridge Electronic Design Ltd | | |
| Software, algorithm | GraphPad Prism 9.1.1 | GraphPad Software | | |

## Study design

Wildtype females were crossed to *Tcf4*$^{STOP/+}$ males to generate wildtype and *Tcf4*$^{STOP/+}$ mice (PTHS model mice). *Tcf4*$^{STOP/+}$ females were crossed to *Cre* transgenic males to conditionally reinstate *Tcf4* expression in a Cre-dependent manner (**Figures 1 and 2**). Neonatal (P1-2) *Tcf4*$^{STOP/+}$ and *Tcf4*$^{+/+}$ mice were randomly assigned to treatment with vehicle or AAV9/PHP.eB-hSyn-Cre at a dose of $3.2 \times 10^{10}$ vg

delivered bilaterally to the cerebral ventricles. Separate cohorts of neonatal $Tcf4^{STOP/+}$ and $Tcf4^{+/+}$ mice received ICV injection of AAV9/PHP.eB-hSyn-GFP at a dose of $8.5 \times 10^9$ vg. All injected mice performed a battery of behavioral tests beginning 2 months of age, spanning a period of 6–7 weeks, in the following order: Open field, object location memory, elevated plus maze, and nest building (*Figure 4* and *Figure 4—figure supplement 1*). All the treated mice underwent electrode implantation 2 weeks after the last behavioral test and recovered from the surgery for at least 7 days. Most treated mice with intact electrode headcaps were subjected to LFP recording. LFP data were acquired for 3 days, 1 hr each day (*Figure 5*). Upon the completion of LFP recording, mice were sacrificed for in situ hybridization (ISH) and qPCR analyses. Half a brain was used for ISH staining, and the other half was used for qPCR measurements (*Figure 6*). All behavioral and LFP data were from two or three biological replicates, and all behavioral experiments were performed only once. All investigators who conducted experiments and analyzed data were blinded to genotype and treatment until completion of the study. Sample size of each behavioral task was determined based on previous publications (*Kennedy et al., 2016*; *Thaxton et al., 2018*).

## Mice

The generation of $Tcf4^{STOP/+}$ knock in mice has been previously described (*Kim et al., 2020*), and this mouse model is available on request. Mice carrying *loxP-GFP-STOP-loxP* allele were maintained on a congenic C57BL/6 J background. The female $Tcf4^{STOP/+}$ mice were mated with heterozygous males from one of three *Cre*-expressing lines: *Neurod6-Cre$^{+/-}$* (*Goebbels et al., 2006*), *Gad2-Cre$^{+/-}$*, *Actb-Cre$^{+/-}$*, *Olig2-Cre$^{+/-}$*, or *Camk2a-Cre$^{+/-}$*. All mice were maintained on a 12:12 light-dark cycle with ad libitum access to food and water. We used male and female littermates at equivalent genotypic ratios. All research procedures using mice were approved by the Institutional Animal Care and Use Committee at the University of North Carolina at Chapel Hill (IACUC protocol# 20–156.0) and Institutional Animal Care and Use Committee at Bates College (IACUC protocol# 21–05) and conformed to National Institutes of Health guidelines.

## Calculation of *Tcf4* expression from public single-cell sequencing data

Single-cell transcriptomic data from the neonatal mouse cortex (*Loo et al., 2019*) and the adult mouse nervous system (*Zeisel et al., 2015*) were obtained from GEO accession GSE123335 and from http://mousebrain.org/downloads.html, respectively. For the neonatal cortex data, the mean and standard error of *Tcf4* expression values were computed across all cells of a given annotated cell type in R and plotted using ggplot2. For adult data, we focused just on cell types annotated as deriving from cortex, amygdala, dentate gyrus, hippocampus, olfactory bulb, cerebellum, and striatum. We then grouped similar cell types into broader classifications. For example, all clusters annotated as glutamatergic (GLU) were renamed as 'Excitatory neuron'. We then computed the mean and standard error of *Tcf4* expression for each broader cell type within each brain region in R and plotted using ggplot2. All code to reproduce the plots for *Figure 2—figure supplement 1* is provided at https://github.com/jeremymsimon/Kim_TCF4, (copy archived at swh:1:rev:63e064495d28f1940e7f9b2b992dbb9dd5263cd9; *Kim, 2022*).

## Adeno-associated viral vector production

To produce AAV9/PHP.eB capsids, a polyethylenimine triple transfection protocol was first performed. Then the product was grown under serum-free conditions and purified through three rounds of CsCl density gradient centrifugation. Purified product was exchanged into storage buffer containing 1 x phosphate-buffered saline (PBS), 5% D-Sorbitol, and 350 mM NaCl. Virus titers (GC/ml) were determined by qPCR targeting the AAV inverted terminal repeats. A codon-optimized *Cre* and GFP cDNA were packaged into AAV9/PHP.eB capsids.

## AAV delivery

P1-2 mouse pups were cryo-anesthetized on ice for about 3 min, then transferred to a chilled stage equipped with a fiber optic light source for transillumination of the lateral ventricles. A 10 ml syringe fitted with a 32-gauge, 0.4-inch-long sterile syringe needle (7803–04, Hamilton) was used to bilaterally deliver 0.5 ml of AAV9/PHP.eB-hSyn-Cre, AAV9/PHP.eB-hSyn-GFP, or vehicle (PBS supplemented with 5% D-Sorbitol and additional 212 mM NaCl) to the ventricles. The addition of Fast Green dye (1 mg/mL)

to the virus solution was used to visualize the injection area. Following injection, pups were warmed on an isothermal heating pad with home-cage nesting material before being returned to their home cages.

## Behavioral testing and analyses

Object location memory: Mice were habituated to an open box, containing a visual cue on one side, without objects for 5 min each day for 3 days. In the following day, mice were trained with two identical objects for 10 min. After 24 hr, mice were placed in the box where one of the objects was relocated to a novel position for 5 min. Video was recorded during each period. Interaction time of a mouse with each object was measured by Ethovision XT 15.0 program. A percentage of the exploration time with the object in a novel position (% in novel location) was calculated as follows: (time exploring novel location)/(time exploring novel location +familiar location) * 100. If total exploration time was less than 2 s, these mice were excluded from the dataset.

Open field (*Figure 1E*): Mice were given a 30-min trial in an open-field chamber (41 × 41 x 30 cm) that was crossed by a grid of photobeams (VersaMax system, AccuScan Instruments). Counts were taken of the number of photobeams broken during the trial in 5 min intervals. Total distance traveled was measured over the course of the 30-min trial.

Open field (*Figures 2B and 4B*): Mice were given a 30-min trial in an open-field chamber (40 × 40 x 30 cm). Mouse movements were recorded with a video camera, and the total distance traveled was measured by Ethovision XT 15.0 program.

Elevated plus maze: The elevated plus maze was constructed to have two open arms and two closed arms; all arms are 20 cm in length and 8 cm in width. The maze was elevated 50 cm above the floor. Mice were placed on the center section and allowed to explore the maze for 5 min. Mouse movements on the maze were recorded with a video camera. Activity levels (time and entry) in open or closed arms were measured by Ethovision XT 15.0 program.

Nest building: Mice were single-housed for a period of 3 days before the start of the assay. On day 1, 10–11 g of compressed extra-thick blot filter paper (1703966, Bio-Rad), cut into 8 evenly sized rectangles, were placed in a cage. In each day, for 4 consecutive days (*Figure 1F*), the amount of paper not incorporated into a nest was weighed. For *Figures 2E and 4E*, additional measurement of nest material was recorded 72 hr after collecting data for 4 consecutive days.

## Surgery and in vivo LFP recording

Mice were anaesthetized by inhalation of 1–1.5% isoflurane (Piramal) in pure $O_2$ during surgery, with 0.25% bupivacaine injected under the scalp for local analgesia and meloxicam (10 mg/kg) subcutaneously administered. Stainless steel bipolar recording electrodes (P1 Technologies) were implanted in the hippocampus (coordinates from bregma: AP = –1.82 mm; ML = 1.5 mm; and DV = –1.2 mm), and ground electrodes were fastened to a stainless-steel screw positioned on the skull above the cerebellum. Dental cement was applied to secure electrode positions. Mice recovered for at least 7 days prior to LFP recording. A tethered system with a commutator (P1 Technologies) was used for recordings, while mice freely moved in their home cages. LFP recordings were amplified (1000 x) using single-channel amplifiers (Grass Technologies), sampled at a rate of 1000 Hz, and filtered at 0.3 Hz high-pass and 100 Hz low-pass filters. All electrical data were digitized with a CED Micro1401 digital acquisition unit (Cambridge Electronic Design Ltd.).

## LFP analysis

Data acquired in Spike2 software (Cambridge Electronic Design Ltd.) were read into Python and further processed with a butter bandpass filter from 1 to 100 Hz to focus on frequencies of our interest. Frequency bands were defined as delta 1–4 Hz, theta 5–8 Hz, beta 13–30 Hz, and gamma 30–50 Hz. Spectral power was analyzed using the Welch's Method, where the power spectral density is estimated by dividing the data into overlapping segments. Sample size ('n') represents the number of mice. For each mouse, we selected the longest continuous period with no movement artifacts for analysis. We averaged processed data obtained across three days. We wrote custom Python scripts to analyze LFP data.

## Tissue preparation

Mice were anesthetized with sodium pentobarbital (60 mg/kg, intraperitoneal injection) before transcranial perfusion with 25 ml of PBS immediately followed by phosphate-buffered 4% paraformaldehyde

(pH 7.4). Brains were postfixed overnight at 4 °C before 24 hr incubations in PBS with 30% sucrose. Brains were sectioned coronally or sagittally at 40 mm using a freezing sliding microtome (Thermo Scientific). Sections were stored at –20 °C in a cryo-preservative solution (45% PBS, 30% ethylene glycol, and 25% glycerol by volume).

## Immunohistochemistry

For immunofluorescent staining, sections were rinsed several times with PBS and PBS containing 0.2% Triton X-100 (PBST). Then sections were blocked with 5% normal goat serum in PBST (NGST) for 1 hr at room temperature (RT). Sections were incubated with primary antibodies diluted in NGST at 4 °C for 24 or 48 hr. After primary antibody incubation, sections were rinsed several times with PBST and then incubated with secondary antibodies for 1 hr at RT. DAPI (4',6-diamidino-2-phenylindole) was added during the secondary antibody incubation at a concentration of 700 ng/ml. All primary and secondary antibodies used for immunohistochemistry are listed in the key resource table.

For chromogenic staining, sections were rinsed with PBS and then incubated with endogenous peroxidases for 5 min in 1.0% $H_2O_2$ in MeOH, followed by PBS rinsing. Sections were washed with PBST several times. Sections were blocked with 5% NGST for 1 hr. Blocked sections were incubated in rabbit anti-GFP in NGST for 24 hr at 4 °C. After sections were incubated in primary antibodies, they were rinsed several times in PBST and then incubated for 1 hr in a biotinylated goat anti-rabbit secondary antibody in NGST. Sections were rinsed in PBST prior to tertiary amplification for 1 hr with the ABC elite avidin-biotin-peroxidase system. To visualize immune complexes amplified by avidin-biotin-peroxidase, sections were incubated for 3 min at RT in 3'3'-diaminobenzidine (DAB) chromogenic substrate (0.02% DAB and 0.01% $H_2O_2$ in PBST).

## In situ hybridization

Brains were extracted and frozen in dry ice. Sections were taken at a thickness of 16 mm. Staining procedure was completed to manufacturer's specifications. RNAscope Fluorescent Multiplex Assay (Advanced Cell Diagnostics), designed to visualize multiple cellular RNA targets in fresh frozen tissues (*Wang et al., 2012*), was used to detect Cre in mouse sections.

## Imaging

Images of brain sections stained by using fluorophore-conjugated secondary antibodies were obtained with Zeiss LSM 710 Confocal Microscope, equipped with ZEN imaging software (Zeiss) and a Nikon Ti2 Eclipse Color and Widefield Microscope (Nikon). Images compared within the same figures were taken using identical imaging parameters. Images within figure panels went through identical modification for brightness and contrast by using Fiji Image J software. Figures were prepared using Adobe Illustrator software (Adobe Systems).

## Quantitative real-time PCR

The neocortical and hippocampal hemispheres were rapidly dissected, snap-frozen with dry ice-ethanol bath, and stored at –80 °C. Total RNAs were extracted using the RNeasy Mini Kit, and reverse transcribed via qScript cDNA SuperMix. The resulting cDNAs constituted the input, and qPCR was performed in a QuantStudio Real-Time PCR system using SYBR green master mix. The specificity of the amplification products was verified by melting curve analysis. All qPCRs were conducted in technical triplicates, and the results were averaged for each sample, normalized to *Actin* expression, and analyzed using the comparative CT method (ΔΔCT). The triplicates are valid only when the standard deviation is smaller than 0.25.

## Western blot

Embryonic day 14.5 brains were sonicated on ice using radioimmunoprecipitation assay buffer (50 mM Tris-HCl pH 8.0, 150 mM NaCl, 1% NP-40, 0.5% Na-deoxycholate, 0.5% SDS, and protease inhibitor cocktail). Tissue homogenates were cleared by centrifugation at 4 °C. Protein concentrations were determined by bicinchoninic acid assay. A total of 20 µg of each sample was separated in 8% sodium dodecyl sulfate–polyacrylamide gel electrophoresis and transferred to polyvinylidene fluoride membranes in ice-cold transfer buffer (25 mM Tris-base, 192 mM glycine, and 20% MeOH). Membranes were blocked in Odyssey Blocking Buffer for 1 hr and then blotted with primary antibodies overnight

at 4 °C and then subsequently, with secondary antibodies for 1 hr at RT. Chemiluminescence reaction was performed using Clarity Western ECL Substrate, which was imaged by an Amersham Imager 680 (GE Healthcare). The signal was quantified using the ImageJ software.

### Statistics

All statistical analyses are illustrated in *Supplementary file 1*. All values are expressed as means ± standard error of the mean (SEM). Asterisks indicate p values: *p < 0.05, **p < 0.005, ***p < 0.001, ****p < 0.0001. GraphPad Prism 9.1.1 software was used for all statistical analyses.

## Acknowledgements

We thank Matthew C Judson for critical readings of the manuscript, Dale Cowley at the UNC Animal Models Core for designing and generating the new mouse model, Klaus-Armin Nave for providing *Neurod6-Cre* mice, and Viktoriya Nikolova for training on behavioral tasks. Funding: This research was supported by the Ann D Bornstein Grant from the Pitt-Hopkins Research Foundation and NINDS grant R01NS114086 to BDP, by the Estonian Research Council grant PUTJD925 to HV, and by the Orphan Disease Center grant MDBR-21–105-Pitt Hopkins to AJK. Microscopy was performed at the UNC Neuroscience Microscopy Core (RRID:SCR_019060), supported in part by funding from the UNC Neuroscience Center Support Grant (NINDS; P30 NS045892); PI: Mark Zylka, and the UNC Intellectual and Developmental Disabilities Research Center Support Grant (NICHD; P50 HD103573; PI: Joseph Piven).

## Additional information

### Funding

| Funder | Grant reference number | Author |
| --- | --- | --- |
| Pitt Hopkins Research Foundation | Ann D. Bornstein Grant | Hyojin Kim<br>Benjamin D Philpot |
| National Institute of Neurological Disorders and Stroke | R01NS114086 | Hyojin Kim<br>Benjamin D Philpot |
| Estonian Research Council | PUTJD925 | Hanna Vihma |
| The Orphan Disease Center | MDBR-21-105-Pitt Hopkins | Andrew J Kennedy |

The funders had no role in study design, data collection and interpretation, or the decision to submit the work for publication.

### Author contributions

Hyojin Kim, Conceptualization, Data curation, Formal analysis, Funding acquisition, Investigation, Methodology, Project administration, Resources, Software, Validation, Visualization, Writing – original draft, Writing – review and editing; Eric B Gao, Adam Draper, Noah C Berens, Data curation, Formal analysis, Writing – review and editing; Hanna Vihma, Data curation, Formal analysis, Funding acquisition; Xinyuan Zhang, Alexandra Higashi-Howard, Data curation, Formal analysis; Kimberly D Ritola, Resources; Jeremy M Simon, Formal analysis, Writing – review and editing; Andrew J Kennedy, Funding acquisition, Resources, Writing – review and editing; Benjamin D Philpot, Conceptualization, Data curation, Funding acquisition, Investigation, Supervision, Writing – original draft, Writing – review and editing

### Author ORCIDs

Hyojin Kim http://orcid.org/0000-0001-8690-5617
Adam Draper http://orcid.org/0000-0003-0788-2088
Noah C Berens http://orcid.org/0000-0002-7792-0142
Hanna Vihma http://orcid.org/0000-0002-6128-636X
Jeremy M Simon http://orcid.org/0000-0003-3906-1663

Benjamin D Philpot [iD] http://orcid.org/0000-0003-2746-9143

### Ethics

All research procedures using mice were approved by the Institutional Animal Care and Use Committee at the University of North Carolina at Chapel Hill (IACUC protocol# 20-156.0) and Institutional Animal Care and Use Committee at Bates College (IACUC protocol# 21-05) and conformed to National Institutes of Health guidelines.

### Decision letter and Author response

Decision letter https://doi.org/10.7554/eLife.72290.sa1
Author response https://doi.org/10.7554/eLife.72290.sa2

## Additional files

### Supplementary files
• MDAR checklist

• Supplementary file 1. Statistics.

### Data availability

Numerical data used to generate all figures are provided in the Figure Source Data files that correspond to figure labels. Single-cell transcriptomic data from the neonatal mouse cortex and the adult mouse nervous system were obtained from GEO accession GSE123335 and from http://mousebrain.org/downloads.html. All code to reproduce the plots is provided at https://github.com/jeremymsimon/Kim_TCF4, (copy archived at swh:1:rev:63e064495d28f1940e7f9b2b992dbb9dd5263cd9).

The following previously published datasets were used:

| Author(s) | Year | Dataset title | Dataset URL | Database and Identifier |
|---|---|---|---|---|
| Loo L, Simon JM, McCoy ES, Niehaus JK, Guo J, Anton ES, Zylka MJ | 2019 | Single-cell transcriptomic analysis of mouse neocortical development | https://www.ncbi.nlm.nih.gov/geo/query/acc.cgi?acc=GSE123335 | NCBI Gene Expression Omnibus, GSE123335 |
| Zeisel A, Muñoz-Manchado AB, Codeluppi S, Lönnerberg P, Manno GL, Juréus A, Marques S, Munguba H, He L, Betsholtz C, Rolny C, Castelo-Branco G, Hjerling-Leffler J | 2015 | Cell types in the mouse cortex and hippocampus revealed by single-cell RNA-seq | https://www.ncbi.nlm.nih.gov/geo/query/acc.cgi?acc=GSE60361 | NCBI Gene Expression Omnibus, GSE60361 |

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
