## [Editor Report]

The manuscript provides a proof of principle concept for rescue of a relatively common neurodevelopmental syndrome. By developing a novel Tcf4 conditional mouse model and demonstrating that PTHS phenotypes could be rescued by Tcf4 reinstatement during early postnatal development in particular cell types, the work sets the stage for future therapeutic efforts.

---

## [Decision Letter]

**Decision letter after peer review:**

Thank you for submitting your article "Rescue of behavioral and electrophysiological phenotypes in a Pitt-Hopkins syndrome mouse model by genetic restoration of Tcf4 expression" for consideration by *eLife*. Your article has been reviewed by 2 peer reviewers, and the evaluation has been overseen by a Reviewing Editor and Catherine Dulac as the Senior Editor. The reviewers have opted to remain anonymous.

The reviewers have discussed their reviews with one another, and the Reviewing Editor has drafted this to help you prepare a revised submission. The reviewers were both enthusiastic about the paper, pending modifications.

Essential revisions:

1) Figure 1: In EPM tests, time spent in center zone should be provided to support the anxiety-like behavior. In open-field tests, time spent in center zone should be also provided.

2) Figure 4: Neonatal ICV injections of Cre did not restore the defects of brain size, but improved the behavioral abnormalities. Wasn't microcephaly responsible for intellectual disability of PHS patients? These should be clearly explained.

3) Figure 6F: This mRNA measurement should be validated by measuring protein levels by quantitative immunoblotting experiments.

4) Consider use of appropriate empty virus control rather than vehicle control for experiments (see below) or document the limitations of the experiments.

*Reviewer #1 (Recommendations for the authors):*

1. Figure 1: Actin-Cre is supposed to be ubiquitously acting beyond brains. Validation of behavioral phenotypes by employing brain-specific Cre driver line(s) should be made to ensure the phenotypes.

2. Figure 1: In EPM tests, time spent in center zone should be provided to support the anxiety-like behavior. In open-field tests, time spent in center zone should be also provided.

3. Figure 2: Behavioral experiments using NEX-Cre and GAD2-Cre should be presented.

4. Figure 2: Tcf4 was reported to be expressed in neurons (glutamergic and GABAergic) and non-neurons (e.g. astrocytes and oligodendrocytes) (Kim et al., 2020). However, interpretation of behavior data using GAD2-Cre (covering diverse types of GABAergic neurons) should be carefully made. Further discussion/interpretation should be provided. As TCF4 loss in the oligodendrocytes was shown to contribute to PHS pathophysiology (Phan et al., 2020), it is very puzzling whether effects of Tcf4 loss in neurons are responsible for the PHS pathogenesis.

5. Figure 2B: The conclusions are not convincing because the bar graphs appear to be very similar in cases for using NEX-Cre and GAD2-Cre mice. More experiments are required to make concrete conclusions.

6. Figure 4: Neonatal ICV injections of Cre did not restore the defects of brain size, but improved the behavioral abnormalities. Wasn't microcephaly responsible for intellectual disability of PHS patients? These should be clearly explained.

7. Figure 5: Why did the authors examine only hippocampal LFPs? As Tcf4 is also expressed in cortex, cortical LFPs should be also measured. In addition, there is a clear decreased trend in β and γ waves. Was anything known about brain wave abnormalities in PHS patients?

8. Figure 6F: This mRNA measurement should be validated by measuring protein levels by quantitative immunoblotting experiments.

*Reviewer #2 (Recommendations for the authors):*

This manuscript describes a systematic effort to determine if restoring the TCF4 gene in Pitt-Hopkins Syndrome is a viable therapeutic concept in a preclinical mouse genetic model. The authors first perform an early gene rescue using a reversible flox-STOP-flox, heterozygous knockdown model. Crossing this model with the early onset Actin-cre transgenic mouse line results in 4 genotypes: 1. TCF4f/+::cre-, TCF4+/+::cre-, 3. TCF4f/+::cre+, 4. TCF4+/+::cre+. The TCF4f/+::cre- mice demonstrate behavioral abnormalities that are rescued in TCF4f/+::cre+ mice with TCF4 gene restored to baseline. Appropriate TCF4+/+ controls are included. Using promoters that express in either excitatory glutamatergic neurons or in inhibitory GABAergic neurons, the authors next demonstrate that while either rescue can lead to reversal of some subsets of behavioral differences, neither cell-type specific rescue results in rescue of all behavioral differences.

To determine if genetic reversal at a later developmental timepoint would be feasible, the authors make use of intracerebroventricular injections of an AAV9 expressing cre under control of a synapsin promoter. Injecting AAV-cre virus into neonatal pups at postnatal day 1, the authors demonstrate partial rescue. Behavioral experiments performed between P60 and P110 demonstrated that the mutant TCF4f/+ mice treated with vehicle demonstrated abnormal behaviors, while the AAV-cre-treated TCF4f/+ mice rescued multiple behaviors. In addition to behavioral differences, the authors recorded local field potentials in the hippocampus; AAV-cre at P1 ICV partially rescued some of these differences in the LFP.

Unfortunately, the early postnatal viral rescue experiments were not as thoroughly controlled as the earlier actin-cre genetic rescue experiments. Rather than use an AAV vector expressing a fluorescent marker protein (GFP) or a scrambled cre, they used only vehicle as a control. This leaves open the unlikely possibility that their virus injection led to rescue of phenotypes simply due to infection of brain cells with the AAV virus itself rather than the cre-mediated gene recombination.. One would have liked to have seen the appropriate negative controls with postnatal virus injections. Another possibility is that some other effect of cre recombinase expression other than genetic effects on their lox-STOP-lox insertion into TCF4 could be responsible for their observed postnatal rescue; this seems very unlikely given their appropriately controlled rescue with the Actin-cre early genetic strategy that was thoroughly controlled for cre effects.

Postmortem studies on the brains of the AAV-cre rescued mice demonstrated cre expression in most, but not all cells of the cortex and hippocampus (particularly the dentate gyrus). TCF4 expression was ~1.3X higher in AAV-cre treated mice compared to vehicle treated mice in the forebrains. Select TCF4 regulated gene expression levels were also at least partially normalized.

Overall, this manuscript suggests that gene replacement therapy that uses the endogenous TCF4 promoter can rescue behavioral differences in a TCF4 heterozygous genetic model of Pitt Hopkins Syndrome. Even incomplete TCF4 expression rescue in a subset of neurons appears to be sufficient in a heterozygous mouse genetic model. The authors suggest that TCF4 normalization may benefit from a relatively "wide therapeutic window". Unfortunately, it is not entirely clear what the authors mean by this. They did not measure the level of TCF4 expression in the individual cells in which cre was expressed. It could be that those cells expressed TCF4 back to WT levels rather than a smaller increase in TCF4 levels. Perhaps the authors were rather suggesting that not all neurons need to have TCF4 gene expression levels restored to typical WT levels, providing a "therapeutic window" with respect to the number of neurons/cells in which TCF4 must be restored to have beneficial effects on behavior.

Certainly, this study does not examine a very broad developmental timecourse, sticking only to very early in development and ~24 hrs after birth. The therapeutic window with respect to timing of rescue during brain development would be of great interest to the scientific community. Knowing that a more fully developed brain might be readily rescued with an AAV-cre virus would certainly broaden the applicability of such gene therapy approaches to the more fully developed brain.

Having a TCF4 gene under control of its endogenous promoter may not be very readily translated to human patients of course. It does prove the principle that under ideal expression conditions (those driven by the native TCF4 promoter), that genetic treatment is theoretically possible. Gene therapy with a completely different or artificial promoter, however, might not lead to the correct levels of TCF4 expression at the correct time during or after brain development. Such an approach could create more problems than it solves. Still, knowing that if ideal expression control is obtained that TCF4 mutations could be rescued through a genetic strategy is clearly an important initial step toward identifying rationally designed gene therapeutic approaches. Time will tell if such an approach is feasible, particularly given the limitations on the size of viral construct packaging to date.

I would clarify exactly what is meant by 'broad therapeutic window'. Does this mean"dose" of virus? Authors don't necessarily know that the level of TCF4 expression in individual cells in the brain is anything less than the WT single cell expression of TCF4. So authors likely simply mean that not all cells of the brain require increased TCF4 expression to rescue behavioral differences. Authors should clarify the meaning of broad temporal therapeutic window during development.

Also recommend that you consider using a more appropriate AAV9-GFP control virus in a few studies to demonstrate lack of reversal with the virus but no cre expression.

---

## [Author Response]

Essential revisions:

1) Figure 1: In EPM tests, time spent in center zone should be provided to support the anxiety-like behavior. In open-field tests, time spent in center zone should be also provided.

We have added both requested center zone analyses (Figure 1).

2) Figure 4: Neonatal ICV injections of Cre did not restore the defects of brain size, but improved the behavioral abnormalities. Wasn't microcephaly responsible for intellectual disability of PHS patients? These should be clearly explained.

The extent to which microcephaly contributes to intellectual disability in Pitt-Hopkins syndrome (PTHS) individuals remains unclear. According to a report of commonly observed dysmorphic and neurobehavioral features in PTHS (Goodspeed et al., 2018), 59% of individuals with PTHS have microcephaly, whereas 100% of them exhibit intellectual disability. Thus, there can be a dissociation between microcephaly and behavioral performance in individuals with PTHS. The basis of microcephaly in PTHS is not yet fully understood, although based on prior work we might expect that alterations in myelination, white matter tract integrity, neuronal migration, and/or corticogenesis might contribute to the observed microcephaly (Peters et al., 2011; Phan et al., 2020; Sherman et al., 2012; Wollnik, 2010). Years of work are needed to determine the basis of the observed microcephaly and to establish the extent to which the microcephaly contributes to various PTHS phenotypes.

Without knowing much about the basis for the observed microcephaly, it is difficult to explain why neuronal *Tcf4* reinstatement failed to restore brain size but could restore behavioral phenotypes. One possibility is that reinstating *Tcf4* expression in oligodendrocytes as well as neurons might be needed to recover microcephaly, although we now show that *Tcf4* reinstatement in oligodendrocytes by itself is not sufficient to improve behavioral phenotypes (Figure 2—figure supplement 3C). A non-mutually exclusive possibility is that microcephaly might only be prevented if there is more widespread reinstatement of *Tcf4* expression across the brain and/or if the reinstatement occurs earlier in life than we tested in the current study. We now discuss these different possibilities in the revision (lines 237-240), and note here that, similar to our study, other preclinical gene therapy studies have shown that behavioral recovery can occur in the absence of significant recovery from microcephaly (Judson et al., 2021).

3) Figure 6F: This mRNA measurement should be validated by measuring protein levels by quantitative immunoblotting experiments.

Quantitative PCR mRNA quantification, in situ and immunocytochemical analyses, and behavioral recovery—each demonstrated in the initial submission—provide compelling evidence that PHP.eB/Cre viral delivery reinstates TCF4 expression in a subset of neurons. Nevertheless, we certainly appreciate the Reviewer’s suggestion to validate the TCF4 reinstatement directly at the protein level, and thus attempted to directly trace evidence of this mosaic TCF4 reinstatement via quantitative immunoblotting as requested. However, as we anticipated, this proved to be technically challenging for multiple reasons. First, while TCF4 levels are high and easily detectable during perinatal periods (e.g. Figure 1C), TCF4 levels are dramatically downregulated during early postnatal life (Phan et al., 2020; Rannals et al., 2016). Thus, endogenous TCF4 protein levels can become more difficult to detect in the brain as mice mature [see, for example, (Phan et al., 2020), Extended Data Figure 1D, which compared TCF4 levels in adult wildtype and *Tcf4* heterozygous mice]. Second, the inherent difficulty of detecting TCF4 is compounded by the mosaic nature of the reinstatement that we achieved in our experiments—maximally, a 50% increase in TCF4 protein distributed among a subset of virally transduced cells. Third, there are multiple TCF4 isoforms—18 (or possibly more) in human (Sepp, Kannike, Eesmaa, Urb, and Timmusk, 2011) and at least 6 isoforms of mouse TCF4 [personal communications, Tõnis Timmusk, and see (Nurm et al., 2021)]—making it very difficult to discern the truncated, mutant TCF4 protein bands from functional, non-truncated TCF4 protein bands on the Western blot gel.

Fortunately, decreasing GFP protein levels serve as a reliable proxy for TCF4 reinstatement. In *Tcf4^STOP/+^* mice, GFP is generated when any isoform of transgenic *Tcf4* is expressed. Due to high-efficiency P2A cleavage, GFP accumulates as a singular protein species within brain cells that can detected with sensitive GFP antibodies. Subsequent to Cre-mediated recombination, the P2A-GFP cassette is excised, and expression of functional TCF4 protein is reinstated. Thus, loss of GFP expression provides a straightforward, albeit indirect readout of TCF4 reinstatement as modeled in our experiments. Accordingly, in *Tcf4^STOP/+^* mice treated with PHP.eB/Cre, we observed a reduction in GFP levels (Author response image 1) that was commensurate with increases in *Tcf4* mRNA levels (Figure 3G).

**Author response image 1. sa2fig1:** Representative Western blot and quantification for GFP and GAPDH loading control protein from P17 mouse brain lysates. Quantification represents data from Vehicle- and PHP.eB/Cre-treated Tcf4Stop/+ mice.

Collectively, our findings lend to an optimistic therapeutic outlook for gene addition strategies in PTHS, in which even mosaic TCF4 reinstatement may provide significant benefits for patients. Other neurodevelopmental disorders, including Rett syndrome, may be similarly amenable to treatment by mosaic gene reinstatement, as evidenced by preclinical studies (Gadalla et al., 2013).

4) Consider use of appropriate empty virus control rather than vehicle control for experiments (see below) or document the limitations of the experiments.

To address the possibility that viral infection itself might affect behavioral phenotypes, we injected AAV9/PHP.eB-hSyn-eGFP into *Tcf4^+/+^* and *Tcf4^STOP/+^* neonates to assess their behavioral outcomes. We found that abnormal behavioral phenotypes persisted in PTHS model mice treated with PHP.eB/GFP (Figure 4—figure supplement 1), suggesting that restoring *Tcf4* expression upon Cre-mediated excision of the STOP cassette drives behavioral rescue in PTHS model mice. We discuss these new data in lines 242 to 248 of the Results.

Reviewer #1 (Recommendations for the authors):

1. Figure 1: Actin-Cre is supposed to be ubiquitously acting beyond brains. Validation of behavioral phenotypes by employing brain-specific Cre driver line(s) should be made to ensure the phenotypes.

We previously showed that CNS-specific heterozygous deletion of *Tcf4* (*Tcf4^flox/+^::Nestin-Cre* mice) produced similar phenotypes as pan-cellular heterozygous deletion of *Tcf4* (*Tcf4^flox/+^::Actin-Cre* mice) in terms of microcephaly, hyperactivity, reduced anxiety, and abnormal spatial learning (Thaxton et al., 2018). Those data suggested that reduction of TCF4 protein in the CNS is sufficient to produce Pitt-Hopkins syndrome-like phenotypes.

Here, we further demonstrated the importance of brain-specific TCF4 re-expression using two different approaches. First, we crossed conditional *Tcf4* reinstatement model mice with inhibitory neuron-specific (*Gad2-Cre*) and excitatory neuron-specific (*Nex-Cre*) Cre-driver mice, each of which showed selective behavioral recovery (Figure 2). Second, we were able to improve behavioral and physiological phenotypes in *Tcf4^STOP/+^* mice by delivering a viral vector encoding Cre under a neuron-specific promoter (i.e., human synapsin, Figures 4-5), thus providing additional evidence for a CNS role of TCF4.

Our previous and current findings together support the importance of CNS expression for most, if not all, measured phenotypes. However, we do not rule out a role for loss of TCF4 expression in the manifestation of key PTHS phenotypes beyond the brain, including gastrointestinal complications (Grubisic, Kennedy, Sweatt, and Parpura, 2015), as we now point out in the revised Discussion (lines 381-384).

2. Figure 1: In EPM tests, time spent in center zone should be provided to support the anxiety-like behavior. In open-field tests, time spent in center zone should be also provided.

We now provide the requested data and have incorporated them into Figure 1. We show increased % of time spent in the center zone of the open field for *Tcf4^STOP/+^* mice compared to their controls (Figure 1E). This result is consistent with our observation that *Tcf4^STOP/+^* mice spent more time in the open arms of the elevated plus maze (Figure 1G), and overall, that they exhibit a decrease in anxiety-like behavior. We did not have the statistical power to resolve differences between the *Tcf4^STOP/+^::Actin-Cre* mice and either the controls or *Tcf4^STOP/+^* mice in the center zone of the open field (Figure 1E).

We also examined center time in the elevated plus maze (Figure 1G), but did not observe any group differences.

3. Figure 2: Behavioral experiments using NEX-Cre and GAD2-Cre should be presented.

We now include the data comparing behavioral performance between *Tcf4^+/+^* and *Gad2-Cre^+/-^* or *Nex-Cre^+/-^* mice (Figure 2—figure supplement 2). There are no statistically significant differences between these groups.

4. Figure 2: Tcf4 was reported to be expressed in neurons (glutamergic and GABAergic) and non-neurons (e.g. astrocytes and oligodendrocytes) (Kim et al., 2020). However, interpretation of behavior data using GAD2-Cre (covering diverse types of GABAergic neurons) should be carefully made. Further discussion/interpretation should be provided.

We appreciate the Reviewer’s comment. Although TCF4 is expressed in multiple inhibitory interneurons subtypes, including parvalbumin, somatostatin, and VIP-expressing neurons (Kim, Berens, Ochandarena, and Philpot, 2020), we observed partial or no improvement in hyperactivity, memory, anxiety phenotype, and innate behavior when *Tcf4* was re-expressed broadly across those inhibitory neuronal types (Figure 2 and Figure 2—figure supplement 3B). Given that we observed improvement in memory, anxiety phenotype, and innate behavior from *Tcf4* reinstatement in excitatory neurons (Figure 2), we can speculate that those behavioral phenotypes might be mainly dependent on the proper function of excitatory neurons.

We now discuss results from *Tcf4^STOP/+^::Gad2-Cre* mice more cautiously in the Discussion (lines 359-369). We acknowledge that Gad2 is expressed broadly across inhibitory cell types, and future studies are needed to understand which inhibitory subclasses contribute to behaviors such as nest building. We also discuss possible experiments that can be performed in the future to better understand the interplay between excitatory and inhibitory neurons in behavioral outputs, highlighting the effect of cell type-specific deletion of *Tcf4* on behavioral phenotypes (lines 367-369).

As TCF4 loss in the oligodendrocytes was shown to contribute to PHS pathophysiology (Phan et al., 2020), it is very puzzling whether effects of Tcf4 loss in neurons are responsible for the PHS pathogenesis.

We are also intrigued by the role of oligodendrocytes in PTHS pathogenesis and were thus compelled to further explore this relationship in our present study. In new experiments, we reinstated *Tcf4* in oligodendrocytes by crossing *Tcf4^STOP/+^* mice with *Olig2-Cre^+/-^* mice, but found that this failed to improve behavioral performance in either the open field or object location memory tasks (Figure 2—figure supplement 3C). These data suggest that reinstatement of *Tcf4* expression in oligodendrocytes is not sufficient to recover behavioral phenotypes in PTHS model mice. However, we cannot rule out that TCF4 reinstatement in oligodendrocytes as well as in neurons might be required for full phenotypic recovery, or that there may be a selective role of TCF4 in oligodendrocytes in other, unmeasured phenotypes. We now briefly discuss these points in the modified Results/Discussion (lines 174-182 and 370-381).

5. Figure 2B: The conclusions are not convincing because the bar graphs appear to be very similar in cases for using NEX-Cre and GAD2-Cre mice. More experiments are required to make concrete conclusions.

We agree. As the Reviewer suggested, we added a pre-determined number of additional mice to these behavioral data, which has strengthened our conclusions (Figure 2).

6. Figure 4: Neonatal ICV injections of Cre did not restore the defects of brain size, but improved the behavioral abnormalities. Wasn't microcephaly responsible for intellectual disability of PHS patients? These should be clearly explained.

See response to essential revisions, response #2.

7. Figure 5: Why did the authors examine only hippocampal LFPs? As Tcf4 is also expressed in cortex, cortical LFPs should be also measured. In addition, there is a clear decreased trend in β and γ waves. Was anything known about brain wave abnormalities in PHS patients?

We decided to examine hippocampal local field potentials for several reasons. First, TCF4 is expressed at particularly high levels in the hippocampus (Jung et al., 2018; Kim et al., 2020). Second, hippocampal synaptic plasticity is disrupted in multiple models of PTHS (Kennedy et al., 2016; Thaxton et al., 2018), suggesting this is an important site of neural dysfunction. Third, hippocampus-dependent behavioral deficits are central features of PTHS mouse models (Kennedy et al., 2016; Thaxton et al., 2018). Finally, because our viral delivery approach consistently transduced neurons in the hippocampus (Figure 3E), we recognized it as an important region in which to monitor for both dysfunction and recovery.

Relatively small-sample studies have described abnormalities in slow wave EEG activity in individuals with PTHS (Amiel et al., 2007; Takano, Lyons, Moyes, Jones, and Schwartz, 2010), and others have noted there is “*no specific EEG pattern….but interestingly the background activity showed diffuse or focal slowing in about half of the patients*” (Matricardi et al., 2022). There is clearly a need for a highly quantitative and appropriately statistically powered study to assess specific EEG deficits and their trajectory across development in PTHS individuals compared to appropriate controls. Likewise, much larger studies than ours, including developmental trajectories, are needed to fully understand EEG abnormalities in PTHS model mice so that these can be more accurately compared to findings in the patient population, but such comparisons seem currently unwarranted or, at least, should be made with caution. The reviewer correctly notes a trend for reduced β and γ EEG rhythms in PTHS model mice, and indeed total EEG power is also reduced. We acknowledge these trends in the revised manuscript, as well as the need for much more extensive EEG analyses in both PTHS model mice and the patient population (lines 267-269).

8. Figure 6F: This mRNA measurement should be validated by measuring protein levels by quantitative immunoblotting experiments.

See response to essential revisions, response #3.

Reviewer #2 (Recommendations for the authors):

This manuscript describes a systematic effort to determine if restoring the TCF4 gene in Pitt-Hopkins Syndrome is a viable therapeutic concept in a preclinical mouse genetic model. The authors first perform an early gene rescue using a reversible flox-STOP-flox, heterozygous knockdown model. Crossing this model with the early onset Actin-cre transgenic mouse line results in 4 genotypes: 1. TCF4f/+::cre-, TCF4+/+::cre-, 3. TCF4f/+::cre+, 4. TCF4+/+::cre+. The TCF4f/+::cre- mice demonstrate behavioral abnormalities that are rescued in TCF4f/+::cre+ mice with TCF4 gene restored to baseline. Appropriate TCF4+/+ controls are included. Using promoters that express in either excitatory glutamatergic neurons or in inhibitory GABAergic neurons, the authors next demonstrate that while either rescue can lead to reversal of some subsets of behavioral differences, neither cell-type specific rescue results in rescue of all behavioral differences.

To determine if genetic reversal at a later developmental timepoint would be feasible, the authors make use of intracerebroventricular injections of an AAV9 expressing cre under control of a synapsin promoter. Injecting AAV-cre virus into neonatal pups at postnatal day 1, the authors demonstrate partial rescue. Behavioral experiments performed between P60 and P110 demonstrated that the mutant TCF4f/+ mice treated with vehicle demonstrated abnormal behaviors, while the AAV-cre-treated TCF4f/+ mice rescued multiple behaviors. In addition to behavioral differences, the authors recorded local field potentials in the hippocampus; AAV-cre at P1 ICV partially rescued some of these differences in the LFP.

Unfortunately, the early postnatal viral rescue experiments were not as thoroughly controlled as the earlier actin-cre genetic rescue experiments. Rather than use an AAV vector expressing a fluorescent marker protein (GFP) or a scrambled cre, they used only vehicle as a control. This leaves open the unlikely possibility that their virus injection led to rescue of phenotypes simply due to infection of brain cells with the AAV virus itself rather than the cre-mediated gene recombination.. One would have liked to have seen the appropriate negative controls with postnatal virus injections. Another possibility is that some other effect of cre recombinase expression other than genetic effects on their lox-STOP-lox insertion into TCF4 could be responsible for their observed postnatal rescue; this seems very unlikely given their appropriately controlled rescue with the Actin-cre early genetic strategy that was thoroughly controlled for cre effects.

We thank the Reviewer for their enthusiasm for our research and for their suggestion to include appropriate negative controls (PHP.eB-hSyn-GFP) for viral-mediated *Tcf4* reinstatement study. We have now done these experiments (see response to essential revisions, response #2).

Postmortem studies on the brains of the AAV-cre rescued mice demonstrated cre expression in most, but not all cells of the cortex and hippocampus (particularly the dentate gyrus). TCF4 expression was ~1.3X higher in AAV-cre treated mice compared to vehicle treated mice in the forebrains. Select TCF4 regulated gene expression levels were also at least partially normalized.

Overall, this manuscript suggests that gene replacement therapy that uses the endogenous TCF4 promoter can rescue behavioral differences in a TCF4 heterozygous genetic model of Pitt Hopkins Syndrome. Even incomplete TCF4 expression rescue in a subset of neurons appears to be sufficient in a heterozygous mouse genetic model. The authors suggest that TCF4 normalization may benefit from a relatively "wide therapeutic window". Unfortunately, it is not entirely clear what the authors mean by this. They did not measure the level of TCF4 expression in the individual cells in which cre was expressed. It could be that those cells expressed TCF4 back to WT levels rather than a smaller increase in TCF4 levels. Perhaps the authors were rather suggesting that not all neurons need to have TCF4 gene expression levels restored to typical WT levels, providing a "therapeutic window" with respect to the number of neurons/cells in which TCF4 must be restored to have beneficial effects on behavior.

We agree that we were loose in our use of the phrase “wide therapeutic window” and have removed this phrase. We agree that a modest increase of *Tcf4* expression level shown in PHP.eB/Cre-treated *Tcf4^STOP/+^* brains appear to be driven by a subset of neurons whose *Tcf4* expression was normalized to its wildtype levels (Figure 6). Accordingly, we have clarified our interpretation of the data in the Results (lines 314-316).

Certainly, this study does not examine a very broad developmental timecourse, sticking only to very early in development and ~24 hrs after birth. The therapeutic window with respect to timing of rescue during brain development would be of great interest to the scientific community. Knowing that a more fully developed brain might be readily rescued with an AAV-cre virus would certainly broaden the applicability of such gene therapy approaches to the more fully developed brain.

We appreciate the Reviewer’s suggestions. Indeed, the primary goal of the next phase of this research program is to define the developmental window for successful intervention. We hope that the reviewer appreciates this is a massive undertaking outside the scope of the current study, as it involves optimizing viral titers, characterizing biodistribution, and performing appropriate controls at multiple ages. This future work has major implications for eventual human clinical trials, as we now highlight in our revised Discussion (lines 333-336).

Having a TCF4 gene under control of its endogenous promoter may not be very readily translated to human patients of course. It does prove the principle that under ideal expression conditions (those driven by the native TCF4 promoter), that genetic treatment is theoretically possible. Gene therapy with a completely different or artificial promoter, however, might not lead to the correct levels of TCF4 expression at the correct time during or after brain development. Such an approach could create more problems than it solves. Still, knowing that if ideal expression control is obtained that TCF4 mutations could be rescued through a genetic strategy is clearly an important initial step toward identifying rationally designed gene therapeutic approaches. Time will tell if such an approach is feasible, particularly given the limitations on the size of viral construct packaging to date.

I would clarify exactly what is meant by 'broad therapeutic window'. Does this mean"dose" of virus? Authors don't necessarily know that the level of TCF4 expression in individual cells in the brain is anything less than the WT single cell expression of TCF4. So authors likely simply mean that not all cells of the brain require increased TCF4 expression to rescue behavioral differences. Authors should clarify the meaning of broad temporal therapeutic window during development.

We apologize for the ambiguous use of a “broad therapeutic window”. We now clarify that there may be an intervention age window, corresponding roughly to less than 2 years of age in humans as roughly extrapolated from the mouse studies, that may effectively treat PTHS-associated phenotypes. Future studies examining juvenile and adult *Tcf4* reinstatement should be able to determine whether PTHS individuals at ages later than approximately 1-2 years old in humans might benefit from genetic therapy. We now clarify these points in the revised Discussion (lines 328-336), and we acknowledge the challenge of basing predictions for eventual human clinical trials on preclinical mouse studies.

Also recommend that you consider using a more appropriate AAV9-GFP control virus in a few studies to demonstrate lack of reversal with the virus but no cre expression.

We added new data showing that delivering AAV9/PHP.eB-GFP control virus does not rescue behavioral phenotypes of *Tcf4^STOP/+^* mice (Figure 4—figure supplement 1). See response #4 to essential revisions.

References

Amiel, J., Rio, M., de Pontual, L., Redon, R., Malan, V., Boddaert, N.,... Colleaux, L. (2007). Mutations in TCF4, encoding a class I basic helix-loop-helix transcription factor, are responsible for Pitt-Hopkins syndrome, a severe epileptic encephalopathy associated with autonomic dysfunction. *Am J Hum Genet, 80*(5), 988-993. doi:10.1086/515582

Dupuis, N., Fafouri, A., Bayot, A., Kumar, M., Lecharpentier, T., Ball, G.,... El Ghouzzi, V. (2015). Dymeclin deficiency causes postnatal microcephaly, hypomyelination and reticulum-to-Golgi trafficking defects in mice and humans. *Hum Mol Genet, 24*(10), 2771-2783. doi:10.1093/hmg/ddv038

Gadalla, K. K., Bailey, M. E., Spike, R. C., Ross, P. D., Woodard, K. T., Kalburgi, S. N.,... Cobb, S. R. (2013). Improved survival and reduced phenotypic severity following AAV9/MECP2 gene transfer to neonatal and juvenile male Mecp2 knockout mice. *Mol Ther, 21*(1), 18-30. doi:10.1038/mt.2012.200

Goodspeed, K., Newsom, C., Morris, M. A., Powell, C., Evans, P., and Golla, S. (2018). Pitt-Hopkins Syndrome: A Review of Current Literature, Clinical Approach, and 23-Patient Case Series. *J Child Neurol, 33*(3), 233-244. doi:10.1177/0883073817750490

Grubisic, V., Kennedy, A. J., Sweatt, J. D., and Parpura, V. (2015). Pitt-Hopkins Mouse Model has Altered Particular Gastrointestinal Transits in vivo. *Autism Res, 8*(5), 629-633. doi:10.1002/aur.1467

Judson, M. C., Shyng, C., Simon, J. M., Davis, C. R., Punt, A. M., Salmon, M. T.,... Philpot, B. D. (2021). Dual-isoform hUBE3A gene transfer improves behavioral and seizure outcomes in Angelman syndrome model mice. *JCI Insight, 6*(20). doi:10.1172/jci.insight.144712

Jung, M., Haberle, B. M., Tschaikowsky, T., Wittmann, M. T., Balta, E. A., Stadler, V. C.,... Lie, D. C. (2018). Analysis of the expression pattern of the schizophrenia-risk and intellectual disability gene TCF4 in the developing and adult brain suggests a role in development and plasticity of cortical and hippocampal neurons. *Mol Autism, 9*, 20. doi:10.1186/s13229-018-0200-1

Kennedy, A. J., Rahn, E. J., Paulukaitis, B. S., Savell, K. E., Kordasiewicz, H. B., Wang, J.,... Sweatt, J. D. (2016). Tcf4 Regulates Synaptic Plasticity, DNA Methylation, and Memory Function. *Cell Rep, 16*(10), 2666-2685. doi:10.1016/j.celrep.2016.08.004

Kim, H., Berens, N. C., Ochandarena, N. E., and Philpot, B. D. (2020). Region and Cell Type Distribution of TCF4 in the Postnatal Mouse Brain. *Front Neuroanat, 14*, 42. doi:10.3389/fnana.2020.00042

Matricardi, S., Bonanni, P., Iapadre, G., Elia, M., Cesaroni, E., Danieli, A.,... Verrotti, A. (2022). Epilepsy, electroclinical features, and long-term outcomes in Pitt-Hopkins syndrome due to pathogenic variants in the TCF4 gene. *Eur J Neurol, 29*(1), 19-25. doi:10.1111/ene.15104

Nurm, K., Sepp, M., Castany-Pladevall, C., Creus-Muncunill, J., Tuvikene, J., Sirp, A.,... Timmusk, T. (2021). Isoform-Specific Reduction of the Basic Helix-Loop-Helix Transcription Factor TCF4 Levels in Huntington's Disease. *eNeuro, 8*(5). doi:10.1523/ENEURO.0197-21.2021

Peters, S. U., Kaufmann, W. E., Bacino, C. A., Anderson, A. W., Adapa, P., Chu, Z.,... Wilde, E. A. (2011). Alterations in white matter pathways in Angelman syndrome. *Dev Med Child Neurol, 53*(4), 361-367. doi:10.1111/j.1469-8749.2010.03838.x

Phan, B. N., Bohlen, J. F., Davis, B. A., Ye, Z., Chen, H. Y., Mayfield, B.,... Maher, B. J. (2020). A myelin-related transcriptomic profile is shared by Pitt-Hopkins syndrome models and human autism spectrum disorder. *Nat Neurosci*. doi:10.1038/s41593-019-0578-x

Pulvers, J. N., Bryk, J., Fish, J. L., Wilsch-Brauninger, M., Arai, Y., Schreier, D.,... Huttner, W. B. (2010). Mutations in mouse Aspm (abnormal spindle-like microcephaly associated) cause not only microcephaly but also major defects in the germline. *Proc Natl Acad Sci U S A, 107*(38), 16595-16600. doi:10.1073/pnas.1010494107

Rannals, M. D., Hamersky, G. R., Page, S. C., Campbell, M. N., Briley, A., Gallo, R. A.,... Maher, B. J. (2016). Psychiatric Risk Gene Transcription Factor 4 Regulates Intrinsic Excitability of Prefrontal Neurons via Repression of SCN10a and KCNQ1. *Neuron, 90*(1), 43-55. doi:10.1016/j.neuron.2016.02.021

Sepp, M., Kannike, K., Eesmaa, A., Urb, M., and Timmusk, T. (2011). Functional diversity of human basic helix-loop-helix transcription factor TCF4 isoforms generated by alternative 5' exon usage and splicing. *PLoS One, 6*(7), e22138. doi:10.1371/journal.pone.0022138

Sherman, D. L., Krols, M., Wu, L. M., Grove, M., Nave, K. A., Gangloff, Y. G., and Brophy, P. J. (2012). Arrest of myelination and reduced axon growth when Schwann cells lack mTOR. *J Neurosci, 32*(5), 1817-1825. doi:10.1523/JNEUROSCI.4814-11.2012

Takano, K., Lyons, M., Moyes, C., Jones, J., and Schwartz, C. E. (2010). Two percent of patients suspected of having Angelman syndrome have TCF4 mutations. *Clin Genet, 78*(3), 282-288. doi:10.1111/j.1399-0004.2010.01380.x

Thaxton, C., Kloth, A. D., Clark, E. P., Moy, S. S., Chitwood, R. A., and Philpot, B. D. (2018). Common Pathophysiology in Multiple Mouse Models of Pitt-Hopkins Syndrome. *J Neurosci, 38*(4), 918-936. doi:10.1523/JNEUROSCI.1305-17.2017

Wollnik, B. (2010). A common mechanism for microcephaly. *Nat Genet, 42*(11), 923-924. doi:10.1038/ng1110-923

Zhou, Z. W., Tapias, A., Bruhn, C., Gruber, R., Sukchev, M., and Wang, Z. Q. (2013). DNA damage response in microcephaly development of MCPH1 mouse model. *DNA Repair (Amst), 12*(8), 645-655. doi:10.1016/j.dnarep.2013.04.017